# MAGICORE: MULTI-AGENT, ITERATIVE, COARSE-TO-FINE REFINEMENT FOR REASONING

## ABSTRACT

Large Language Models' (LLM) reasoning can be improved using test-time aggregation strategies, i.e., generating multiple samples for each problem and aggregating over them to find a better answer. While these improve performance, they often reach a saturation point beyond which additional samples provide no return. Refinement offers an alternative by using model-generated feedback to improve answer quality. However, refinement introduces three key challenges: **(1) Excessive refinement:** Uniformly refining all instances can cause over-correction and reduce overall performance. **(2) Inability to localize and address errors:** LLMs have a limited ability to self-correct and struggle to identify and correct their own mistakes in a targeted way. **(3) Insufficient refinement:** Deciding how many iterations of refinement are needed is non-trivial, and stopping too soon could leave errors unaddressed. To tackle these issues, we propose MAGICORE, a framework for **M**ulti **Ag**ent **I**teration for **Co**arse-to-fine **Re**finement. MAGICORE aims to avoid excessive refinement by categorizing problems as easy or hard, solving easy problems with coarse-grained aggregation, and solving hard ones with fine-grained and iterative multi-agent refinement. To enable more granular error localization, we incorporate external step-wise reward model (RM) scores. To ensure effective refinement, we employ a multi-agent loop with three agents: the *Solver*, the *Reviewer* (which generates targeted feedback based on step-wise RM scores) and the *Refiner* (which incorporates feedback and generates new solutions). To ensure sufficient refinement, we re-evaluate updated solutions, iteratively initiating further rounds of multi-agent refinement. We evaluate MAGICORE on Llama-3-8B and GPT-3.5 and show its effectiveness across five math reasoning datasets, with consistent gains for all datasets and models. Even one iteration of MAGICORE beats Self-Consistency by $3.4\%$, Best-of-$k$ by 3.2%, and Self-Refine by $4.0\%$ while using less than $50\%$ of the samples. Unlike iterative refinement with baseline methods, MAGICORE continues to improve with more iterations. Finally, our ablations highlight the importance of MAGICORE's use of RMs and multi-agent communication. [1]

## 1 INTRODUCTION

Imagine a person taking a math exam with problems of varying difficulty; even before answering any question, an effective exam-taker might first distinguish between easier and more challenging problems, deciding how much effort to budget for each one (Son & Metcalfe, 2000; Dodeen, 2015). To maximize their score, the student would likely spend minimal time on the easy problems and focus more on the harder ones, refining their answers where needed. Misallocating effort could not only waste time but even lower their score, as overthinking simple problems might lead to mistakes; similarly failing to dedicate enough thought to hard problems will lead to poor results. For Large Language Models (LLMs) performing reasoning tasks, several test-time approaches dedicate more computation through aggregation to improve performance. These approaches sample multiple solutions to the same question and aggregate over the resulting answers (e.g. Self-Consistency (SC; Wang et al., 2022), Best-of-$k$ sampling (Lightman et al., 2023; Sun et al., 2024; Wang et al., 2023)). However, applying aggregation strategies uniformly on all samples may waste computation on easier problems, where additional samples might not change the answer (Aggarwal et al., 2023). Refinement

---

[1]Code is provided with the supplementary materials.

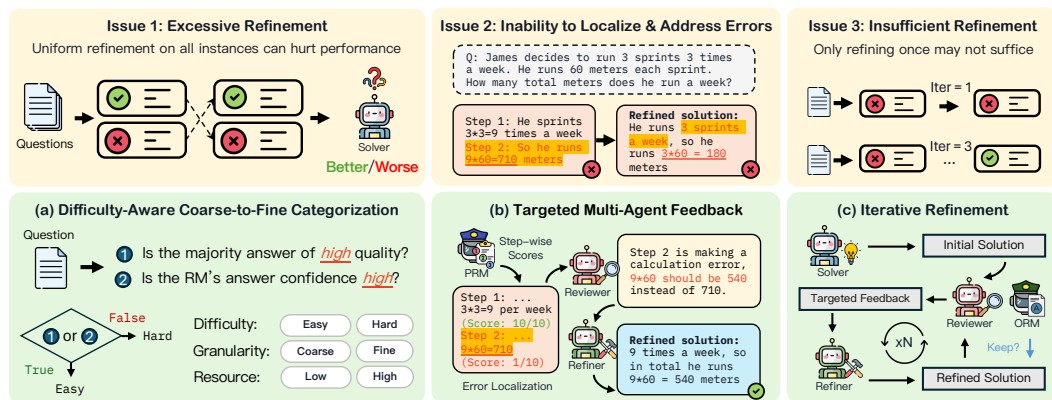

Figure 1: Three main issues in refinement: *(1) Excessive refinement:* refinement risks over-correcting, turning correct solutions into incorrect ones. *(2) Inability to localize and address errors:* LLMs struggle with finding and correcting errors in a targeted way. *(3) Insufficient refinement:* an effective refinement should gradually make the output better, but stopping too soon could leave errors uncorrected. To address these issues, MAGICORE adapts resource allocation based on problem difficulty and refines only when encountering hard problems, adopting a coarse-grained aggregation approach for easy ones. For hard cases requiring refinement, we employ a multi-agent setup that iteratively reviews and refines the solutions based on targeted feedback generated with step-wise PRM scores.

– where solutions are instead critiqued and improved upon during resampling – offers a possible avenue for breaking out of the aggregation rut. This mirrors human reasoning, where feedback can be used to improve answers (rather than simply retrying the same problem), often in an iterative fashion. For example, a teacher might improve a student's understanding by providing multiple rounds of feedback on a test (Pan & Sana, 2021; Roediger & Karpicke, 2006; Wojcikowski & Kirk, 2013).

While refinement seems promising, it faces three key challenges that current work has yet to fully address (outlined in Fig. 1)[2]: **(1) Excessive refinement**: the LLM must know when to refine and when not to. While refinement can help on hard problems, uniformly refining all instances can cause over-refinement, where solutions that were already correct before refinement are "overthought" and flipped to incorrect, reducing the overall performance (Huang et al., 2024; Shridhar et al., 2024; Stechly et al., 2024). **(2) Inability to localize and address errors**: LLMs struggle to identify their own mistakes (i.e., steps needing refinement) and struggle to correct them without explicit instructions. **(3) Insufficient Refinement**: deciding how many iterations of refinement are needed is non-trivial – some cases may require only one round, while others need more, and stopping early could leave errors unaddressed, i.e., hard problems might be "underthought" by a single refinement iteration.

To address these issues and unlock performance gains via refinement, we propose a unified solution, MAGICORE, a framework for **M**ulti **Ag**ent **I**teration for **Co**arse-to-fine **Re**finement. As shown in Fig. 1, our approach leverages external global and local Reward Models (RMs) that enhance both aggregation and refinement processes. To tackle the first issue (excessive refinement), we perform *selective refinement* (see Fig. 1(a)): we start by generating multiple reasoning chains from the LLM and scoring them with off-the-shelf RMs, using the entropy of the final answer distribution to classify examples as easy or hard. This classification allocates resources more efficiently and avoids over-correction by refining only hard samples. Given LLMs' general inability to localize errors (Tyen et al., 2024), we leverage *step-by-step RM scores* to help the LLM pinpoint low-scoring steps (which are likely to be incorrect); this process is shown in Fig. 1(b). Moreover, to help LLMs refine effectively once the errors have been localized, we propose a *multi-agent system* consisting of three agents: the Solver, the Reviewer, and the Refiner. For each problem, the Solver generates a reasoning chain, the Reviewer gives targeted feedback based on step-by-step RM scores, and the Refiner improves the solutions using this feedback. Finally, to address the issue of insufficient refinement, we *iterate the review-refine process*, checking the quality and the entropy of the final answer at each iteration as a stopping criterion (cf. Fig. 1(c)). This makes the communication between the Reviewer and

---

[2]These issues have varying degree of impact on the refinement ability, which we explore later in Table 5.

Refiner agents *bidirectional*, as the refinement process is conditioned on feedback from the review, and subsequent reviews are conditioned on the refined outputs from prior iterations.

We evaluate MAGICoRe on five math reasoning datasets (GSM8K (Cobbe et al., 2021), SVAMP (Patel et al., 2021), MATH (Hendrycks et al., 2021b), MMLU (Hendrycks et al., 2021a) and SAT (Zhong et al., 2023)) with two LLMs: Llama-3-8B and GPT-3.5. Notably, our results show that across datasets and models, we obtain consistent improvements over all aggregation and refinement baselines. Specifically, even just one iteration of MAGICoRe on Llama-3-8B already outperforms Best-of-$k$ sampling (Lightman et al., 2023) by 3.2% and Self-Consistency (Wang et al., 2022) by 3.4%, while using roughly half the samples. MAGICoRe also outperforms a combination of Self-Refine (Madaan et al., 2023) and Self-Consistency by 4.0% and these trends also hold true for GPT-3.5. We demonstrate that MAGICoRe effectively decides when to use refinement and when *not* to, leading to up to a 6.4% improvement over the strongest Best-of-$k$ baseline on MATH (Hendrycks et al., 2021b), whereas uniformly applying refinement to all samples can result in a *drop* of 5.2%, highlighting the key role played by selective refinement. Notably, MAGICoRe further improves with more iterations of refinement, while the baselines stagnate. Our ablations also underscore the benefits of external scoring models (as opposed to LLM self-verification) and the importance of a multi-agent framing. In summary, our contributions and findings are:

- We propose MAGICoRe, a multi-agent coarse-to-fine framework that adaptively uses targeted feedback to outperform refinement methods such as Self-Refine, as well as aggregation methods such as Best-of-$k$ and Self-Consistency, at all sample budgets across five datasets and two LLMs.
- Our difficulty-aware coarse-to-fine resource allocation surpasses $k$-way SC while using $8\times$ fewer samples, and our difficulty classifications correlate with human judgments.
- We show the necessity of incorporating RMs for refinement: using LLM alone is less effective in determining when refinement is needed compared to trained off-the-shelf RMs. Also, using step-wise scores to guide feedback throughout the refinement process results in superior performance.
- Lastly, we validate the importance of iterative refinement: while our refinement baselines do not show improvement over iterations, MAGICoRe consistently improves with additional iterations.

## 2 MAGICoRe: An Adaptive **Co**arse-to-**Fi**ne **Re**finement Framework

MAGICoRe is an adaptive framework that aims to improve both the performance and efficiency of multi-step reasoning with LLMs by intelligently applying test-time aggregation and refinement. MAGICoRe consists of three intertwined components, resulting in a unified solution to the three issues discussed in Section 1: Excessive refinement, the inability of LLMs to localize and address their own errors, and insufficient refinement: (1) MAGICoRe avoids excessive refinement by categorizing problem difficulty as easy or hard, using coarse-grained aggregation for easy problems and fine-grained, iterative multi-agent refinement for hard ones. (2) To improve error localization, we incorporate step-wise scores from an external reward model, providing targeted feedback and thereby helping the LLM to identify problematic steps. To enable effective refinement where the feedback is properly incorporated, we employ a multi-agent setup with three agents: the *Solver*, the *Reviewer*, which generates targeted feedback based on step-wise RM scores, and the *Refiner*, which incorporates feedback to generate new solutions. (3) To ensure sufficient refinement, the refined solutions are re-evaluated, iteratively initiating further rounds of multi-agent refinement if needed, and making the communication between agents bidirectional. Below we introduce MAGICoRe in detail.

### 2.1 Overall MAGICoRe Framework

**Background.** We incorporate three models: (1) an LLM with three roles: the Solver, the Reviewer, and the Refiner for generating initial solutions, targeted feedback, and refinement, respectively (more details in Section 2.3), (2) an Outcome Reward Model (ORM; Cai et al., 2024) for generating global, solution-level correctness score, and (3) a Process Reward Model (PRM; Wang et al., 2023) for generating local step-by-step correctness scores. The ORM and PRM perform both independent and joint tasks. The ORM's global correctness score is crucial for deciding which refined samples to retain during iterative refinement, and the PRM's local correctness score is used for generating targeted feedback. Both the ORM and PRM contribute to (1) assessing problem difficulty and (2) Weighted Self-Consistency (Li et al., 2023) for final answer selection (see Appendix B for details).

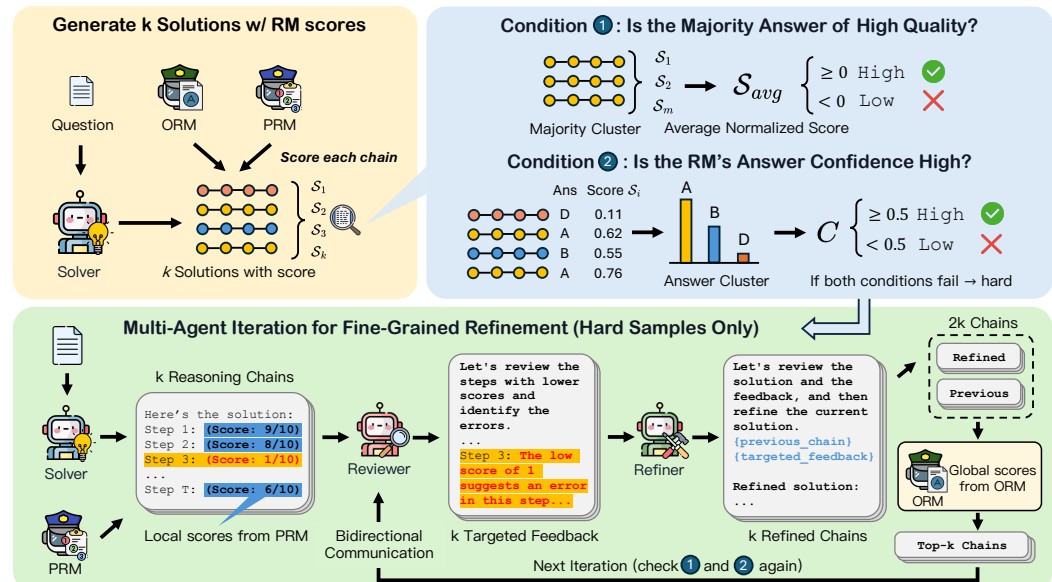

Figure 2: MAGICORE first generates $k$ reasoning chains for each problem and obtains solution-level scores from both the ORM and PRM. The problem's difficulty is then classified based on two conditions: (1) the quality of the majority answer and (2) the RM's answer confidence. A problem is considered hard when the majority answer receives a low average RM score *and* the answer distribution is flat, indicating that no single answer is substantially better than others (i.e., low confidence). For these hard problems, we employ a multi-agent setup with three agents: Solver, Reviewer and Refiner. The Solver generates $k$ reasoning chains, passing them to the PRM for step-wise scoring to pinpoint errors. The Reviewer turns scores into targeted feedback, and the Refiner improves the $k$ solutions using the Reviewer's feedback. This review-refine communication cycle is repeated until either of the two conditions passes, or a maximum iteration is reached.

**Overview.** A high-level overview of MAGICORE is presented in Fig. 2. The process begins with the Solver generating $k$ reasoning chains for each problem, followed by the ORM and PRM providing solution-level scores (as illustrated in the top-left of Fig. 2). Next, the input problem's difficulty is classified based on two criteria (top-right of Fig. 2): (1) the quality of the majority answer and (2) the RMs' answer confidence. Refinement is initiated only when the problem is deemed difficult, which occurs when the majority answer receives a low average RM score *and* the answer distribution is flat – indicating no single answer is significantly better than the others (i.e., low confidence). For these hard samples requiring refinement, we employ a multi-agent setup with three agents: the Solver, the Reviewer, and the Refiner (bottom of Fig. 2). The Reviewer uses the $k$ reasoning chains generated by the Solver, along with the step-wise scores from the PRM, to provide targeted feedback. The Refiner then enhances the $k$ solutions based on this feedback. This bidirectional communication between the Reviewer and Refiner can continue for multiple iterations. At the end of each iteration, the ORM assigns a global correctness score to each chain, determining which top $k$ solutions to retain, with the final answer being aggregated from these top $k$ solutions.

## 2.2 CLASSIFYING PROBLEM DIFFICULTY AND ASSIGNING TO COARSE OR FINE METHODS

We categorize each problem's difficulty as easy or hard using the following conditions (cf. Fig. 2). **Condition 1: Is the Majority Answer of High Quality?** The Solver generates $k$ solutions for the input question and we group them by their final answers. From the largest cluster of solutions, we calculate the average RM score and normalize it by the average score across all solutions, denoted as $\mathcal{S}_{avg}$. After normalization, if $\mathcal{S}_{avg} \geq 0$, Condition 1 is *true*, meaning the majority answer is already high-quality, and hence no refinement is needed. Otherwise, Condition 1 is *false*, suggesting even the most frequent answer is poor, and refinement may help. We evaluate the quality of the majority answer using both ORM and PRM separately but with the same procedure, i.e., if any of the RMs gives the majority answer a low score, Condition 1 will be *false* (see Appendix C for more details). **Condition 2: Is Reward Models' Answer Confidence High?** We check if the RMs are confident in

any single answer. Confidence is determined by how concentrated the answer distribution is (i.e., if one cluster stands out). A concentrated distribution indicates high confidence, while a spread-out distribution shows low confidence. The answer distribution is formed by (1) the frequency of each unique answer and (2) the total RM score of each answer cluster. That is, we weigh each answer cluster by its aggregated RM score, similar to the approach used in Weighted Self-Consistency (Li et al., 2023). We calculate confidence (denoted as $C$) using the entropy of the answer clusters. If $C \geq 0.5$, Condition 2 is *true*, meaning RMs are confident in one answer. Otherwise, Condition 2 is false, indicating uncertainty and the need for further refinement. Note that $0.5$ is a natural threshold requiring no tuning since we pass the entropy to a sigmoid function; further details can be found in Appendix C. Again, we evaluate both ORM and PRM's answer confidence separately, each following the same procedure (i.e., if any of the RMs has low confidence, Condition 2 is *false*).

**Coarse-to-Fine Decision.** If *either* of the conditions is met (the quality of the answer is high *or* the RMs are confident on an answer), an instance is delegated to the easy – or coarse-grained – method. Conversely, if *both* conditions are not satisfied, the instance is delegated to the hard – or fine-grained – method (as described in Section 2.3). For easy instances, we apply the coarse-grained method: Weighted Self-Consistency (Li et al., 2023), using the sum of the solution-level scores generated by both ORM and PRM[3]. Therefore, the coarse-grained method does *not* involve refinement.

## 2.3 FINE-GRAINED MULTI-AGENT ITERATIVE REFINEMENT

Our multi-agent setup has three agents: (1) the *Solver*, which generates the initial solution (2) the *Reviewer*, which takes PRM scores and a reasoning chain as input, and generates targeted feedback that pinpoints the errors within the chain, and (3) the *Refiner*, which takes the feedback generated by the Reviewer to refine the previous chain.

**Solver generates $k$ solutions.** Recall that in Section 2.2, we assess problem difficulty using $k$ generated solutions. Even for easy instances where refinement is not needed, generating $k$ solutions and aggregating them improves the performance (Wang et al., 2022). For hard instances, refinement is applied on top of the $k$ solutions simultaneously. Thus, whether refinement is needed or not, generating $k$ solutions is necessary. Consequently, MAGICORE begins by generating $k$ solutions to a given problem. Implementationally, this is done when assessing problem difficulty (see top-left of Fig. 2, where the solutions are generated and scored). When a problem is classified as easy , we aggregate the $k$ solution without refinement. When a problem is classified as hard (i.e., would benefit from refinement and does not fall in the purview of "Issue 1"), we can directly re-use the $k$ solutions *already generated* by the Solver.

**Reviewer generates targeted feedback.** As we pointed to in Fig. 1, even when we know refinement is warranted, LLMs (like the Reviewer agent) may struggle to isolate and localize errors in their own responses (Huang et al., 2024), which we highlight as "Issue 2" of refinement (Inability to Localize and Address Errors). Therefore, to assist the Reviewer in effectively localizing errors and generating useful feedback, we supply the Reviewer with external step-wise PRM scores for each step of the solution. We append these scores to the end of each step (see the bottom-left of Fig. 2, where scores are assigned), and pass the result to the Reviewer. The goal of the Reviewer is to incorporate step-wise correctness scores from the PRM to generate actionable feedback. That is, it takes a chain with the PRM scores as input, and is prompted to identify problematic steps that need refinement and possible ways to fix them. We use a 1-shot prompt to guide the Reviewer how to generate targeted feedback, and the prompt is shown in Appendix G.

**Refiner improves solutions based on targeted Reviewer feedback.** The Refiner then uses the targeted feedback generated by the Reviewer to refine the reasoning chain that was generated by the Solver. Inspired by the finding that LLM can resolve errors when explicitly pointed out (Tyen et al., 2024), and given that the Reviewer already indicates *which* steps of the reasoning chain should be refined, the Refiner agent can focus exclusively on *how the step should be modified* so as to resolve or fix the error based on feedback from the Reviewer. Here we again use a 1-shot prompt to teach the Refiner how to fix the error and improve a reasoning chain based on targeted feedback, and the prompt for the Refiner is shown in Appendix G.

**Iterating the refinement process for bidirectional communication.** For some hard instances, one round of refinement may be insufficient, as the Reviewer may have generated some irrelevant

---

[3]In Table 10 we show that this combination outperforms using the ORM or PRM only.

feedback or the Refiner may not have fixed the highlighted step adequately ("Issue 3" in Fig. 1). Thus, the Reviewer and the Refiner need to collaborate with each other over the course of multiple refinement iterations. This creates a bidirectional communication cycle between the Reviewer and the Refiner, with outputs from the Refiner going back to the Reviewer in the next round. To prevent excessive refinement, we re-evaluate the two conditions described in Section 2.2 in each iteration. The refinement continues until (a) one of the conditions is met, or (b) a predefined maximum number of iterations is reached. This process is illustrated in Fig. 2 (bottom panel) by the arrow going from the Refiner output back to the Reviewer, which shows both conditions being re-assessed.

**Final answer selection.** The refinement process described above operates on all $k$ chains simultaneously, producing $k$ refined chains in each iteration. At the end of each iteration, we use the ORM to assess whether the refined solution has improved based on its global correctness score. In other words, by the end of each iteration, we have $2k$ reasoning chains – $k$ initial and $k$ refined – but retain only the top $k$ based on their global ORM scores. Here we choose to base the decision on the ORM score because the targeted feedback is generated with PRM's step-wise scores, so selecting the solution via another scoring model avoids overfitting. Finally, the answer is selected using Weighted Self-Consistency over these retained top $k$ chains, at the end of each iteration.

**Key Improvements.** By leveraging external RMs and an iterative multi-agent setup, our method (1) avoids excessive refinements by only refining hard samples. (2) Enhances error localization and resolution by using PRM's step-wise scoring to offer targeted feedback, enabling effective communication between the Reviewer and the Refiner. (3) Ensures thorough refinement by re-evaluating refined solutions, and initiating additional rounds of refinement when necessary.

## 3 EXPERIMENTAL SETUP

**Implementation Details.** We develop MAGICORE with `Llama3-8B-Instruct` (AI@Meta, 2024) and `GPT-3.5-Turbo` (OpenAI, 2022) as the base LLMs. Based on their strong performance on standard reward modeling evaluations (Lambert et al., 2024), we choose InternLM-7B (Cai et al., 2024) as the ORM, and Math-Shepherd-7B (Wang et al., 2023) as the PRM for computing the RM scores. By default, we sample $k = 40$ reasoning chains in each iteration for MAGICORE, and the decoding temperature is set to $0.8$. The maximum number of iterations is set to 3, with additional analysis in Fig. 3 and Table 12. We compare against different categories of baselines, as follows.

- **Vanilla Prompting.** The first baseline as a reference is zero-shot Chain-of-Thought (Wei et al., 2022); note that this only generates one reasoning chain per question without aggregation.
- **Iterative Prompting.** Next, we compare MAGICORE to an iterative prompting method, **Self-Refine (SR)** (Madaan et al., 2023), which refines the initial CoT answer via iteratively prompting the LLM to generate feedback and refine the previous output accordingly.
- **Aggregation-based Methods.** The third category generates multiple samples for each question. Here, we sample $k$ solutions from the same LLM and select the final answer either via $k$-way **Self-Consistency (SC)** (Wang et al., 2022) or according to the highest ORM score (**Best-of-$k$**) (Lightman et al., 2023; Sun et al., 2024; Wang et al., 2023). Note that we compare with $k = 120$ for these aggregation baselines, but in MAGICORE, only the hard instances require more than $40$ samples (with up to $3 \times 40 = 120$ samples used in cases needing three iterations of refinement).
- **Iterative Baseline with Aggregation.** To enable a fair comparison, we also report a stronger version of self-refine by combining **Self-Refine and Self-Consistency (SR+SC)**, i.e., a baseline that is iterative, refines, and aggregates. Specifically, this baseline applies Self-Refine for $k$ samples in parallel, and the final answer is derived by aggregating the $k$ refined solutions.

**Datasets.** We evaluate MAGICORE extensively on five math reasoning datasets. The first class of datasets are math word problems: **GSM8K** (Cobbe et al., 2021), **SVAMP** (Patel et al., 2021), and **MATH** (Hendrycks et al., 2021b). GSM8K and SVAMP consist of grade school-level math problems, with 1312 and 1000 test samples. MATH comprises high-school math competitions spanning diverse topics and a total of 5000 problems. Following previous works (Lightman et al., 2023; Wang et al., 2023), we evaluate MATH performance on a representative subset of 500 samples. We also evaluate on mathematics splits of general benchmarks that test language models' world knowledge and problem-solving abilities over various subjects such as **MMLU-Math** (Hendrycks et al., 2021a; Yue et al., 2024) and **SAT** (Zhong et al., 2023) with 974 and 220 test instances respectively.

Table 1: Performance comparison of different methods and models. The (+x%) is compared with the strongest baseline (Best-of-$k$) highlighted in blue. Across both models and datasets, MAGICORE provides consistent improvements. Notably, MAGICORE already surpasses all baselines after the *first iteration* of refinement, even when the baselines use a larger sample size ($k = 120$).

| | MMLU | MATH | SVAMP | GSM8K | SAT | Avg. |
|---|---|---|---|---|---|---|
| | | | Llama3-8B-Instruct | | | |
| Zero-shot CoT | 50.4 | 24.2 | 72.4 | 80.1 | 58.2 | 57.1 |
| Self-Refine (Best Iter) | 49.8 | 24.0 | 72.6 | 79.6 | 59.6 | 57.1 |
| Best-of-$k$ ($k = 120$) | 62.6 | 41.4 | 88.7 | 90.1 | 72.4 | 71.0 |
| $k$-way SC ($k = 120$) | 63.0 | 40.6 | 89.8 | 90.3 | 70.5 | 70.8 |
| Self-Refine + $k$-way SC (Best Iter) | 62.3 | 41.0 | 89.2 | 90.3 | 68.0 | 70.2 |
| MAGICORE (Iter=1) | 67.3 | 46.0 | **91.4** | 91.1 | 75.0 | 74.2 |
| MAGICORE (Iter=2) | 68.4 | 47.2 | 91.1 | **92.3** | 76.4 | 75.1 |
| MAGICORE (Iter=3) | **68.9** (+5.6%) | **47.8** (+5.2%) | 91.3 (+1.7%) | 91.6 (+1.3%) | **78.2** (+5.8%) | **75.6** (+4.3%) |
| | | | GPT-3.5-Turbo | | | |
| Zero-shot CoT | 62.5 | 37.2 | 78.1 | 78.5 | 76.8 | 66.6 |
| Self-Refine (Best Iter) | 61.1 | 37.4 | 77.9 | 78.4 | 77.1 | 66.4 |
| Best-of-$k$ ($k = 120$) | 70.1 | 50.6 | 87.7 | 90.5 | 87.8 | 77.3 |
| $k$-way SC ($k = 120$) | 70.4 | 51.2 | 86.9 | 89.8 | 87.6 | 77.1 |
| Self-Refine + $k$-way SC (Best Iter) | 70.1 | 49.4 | 88.1 | 88.1 | 84.5 | 76.0 |
| MAGICORE (Iter=1) | **73.7** | 57.2 | 89.4 | 91.1 | 90.1 | 80.3 |
| MAGICORE (Iter=2) | 73.3 | 57.8 | **90.1** | 91.1 | **90.9** | 80.6 |
| MAGICORE (Iter=3) | 73.6 (+3.5%) | **58.6** (+8.0%) | **90.1** (+2.4%) | **91.4** (+0.9%) | **90.9** (+3.1%) | **80.9** (+3.6%) |

## 4 RESULTS AND ANALYSIS

### 4.1 MAIN RESULTS

**MAGICORE outperforms all baselines at the first iteration.** We present our main results in Table 1. First, one iteration of MAGICORE already outperforms all baselines. Compared to aggregation-based methods, which generate multiple responses for each problem without refinement, we observe that MAGICORE improves over Best-of-120 by 3.2% (absolute) averaged across the five datasets on Llama-3-8B, despite using $2\times$ fewer samples. Note that our method's first iteration only involves 40 samples for easy problems and 40 refined chains for the *subset* of hard problems, making our $k = 55$ on average. When compared to 120-way SC, our method shows an even greater average improvement of 3.3% on Llama-3-8B and 3.2% on GPT-3.5. Turning to refinement-based methods, we compare against Self-Refine (SR; Madaan et al., 2023) without aggregation, where each iteration operates on a single instance. Note that we run refinement-based baselines with up to 5 iterations and only report the best in Table 1 (denoted as "Best Iter"), leaving a more detailed comparison in Fig. 3 and Table 12. On average, MAGICORE shows 17.1% and 13.5% improvements over SR for Llama-3-8B and GPT-3.5. As SR alone is a weaker baseline without aggregating multiple samples, we also compare to SR + SC. Even when compared to SR + SC's best iteration, MAGICORE outperforms it by 5.4% (Llama-3-8B) and 4.9% (GPT-3.5) on average. This can be attributed to the fact that MAGICORE can adaptively address challenging instances with targeted refinement.

**MAGICORE continues to improve with more iterations.** While MAGICORE already beats *all* baselines after the very first iteration, in Table 1, we also observe a clear upward trend in performance as the number of iterations increases. We illustrate this further in Fig. 3, which presents the accuracy across successive iterations. Our comparison includes Best-of-$k$ and SR + SC with $k = 40$, with accuracy averaged across five datasets. We find that while SR + SC fluctuates around the same range of 70%, MAGICORE continues to improve and stabilize at the third iteration with 75.6% accuracy (with a noticeable 1.4% improvement compared to the first iteration). This highlights the importance of our iterative refinement and the ability to overcome insufficient refinement for hard instances.

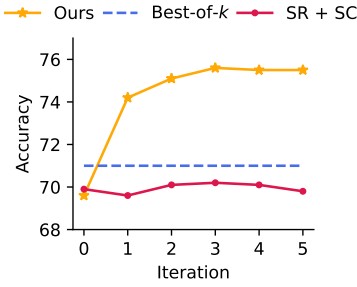

Figure 3: Comparison of our method with baselines across iterations, averaged across five datasets. See Table 12 for full results.

**MAGICORE outperforms aggregation-based methods despite using less computation.** In Fig. 4, we further compare the cost and performance of MAGICORE with Best-of-$k$ and SC on MATH and MMLU using Llama-3-8B, studying how performance changes as we increase the number of reasoning chains generated per question $k$. Note that the $k$ for baselines are the actual sample size, whereas, in our method, we add $k$ more samples in each iteration, resulting in slightly more sample size on average. However, the trend in Fig. 4 shows that MAGICORE consistently outperforms $k$-way SC and Best-of-$k$ at any given $k$. Moreover, while SC saturates and stops improving at around $k = 40$, MAGICORE continues to improve with increasing $k$. Notably, MAGICORE with $k = 15$ already performs better than Best-of-120 and 120-

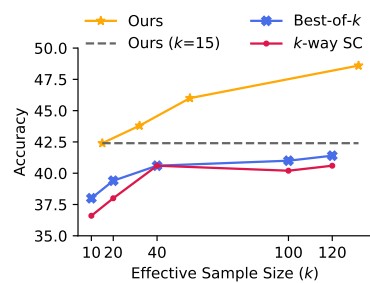

Figure 4: Comparison of MAGICORE, $k$-way SC, and Best-of-$k$ with different $k$ on MATH.

way SC, highlighting the cost-effectiveness of our method. These findings also align with previous research (Chen et al., 2024b) that SC exhibits saturation: there is no meaningful improvement when the number of reasoning chains exceeds $40$, and we even see a performance drop on MATH. In contrast, MAGICORE shows noticeable performance gains with higher $k$, suggesting that MAGICORE possesses a higher ceiling in comparison to SC.

## 4.2 ADDITIONAL ANALYSES OF MAGICORE

**Selective refinement avoids over-correcting and improves overall performance.** In Section 1, we noted that excessive refinement could potentially hurt performance by flipping correct answers to incorrect ones. Here, we provide a quantitative analysis of this claim. Recall that we have two methods: coarse aggregation (Weighted SC) and fine refinement (multi-agent iteration) which we apply *selectively* depending on predicted problem difficulty (c.f. Section 2.2). In Table 2, we measure the performance of each method when applied uniformly to

Table 2: Comparison when uniformly adopting aggregation (i.e., Weighted SC) or refinement to *all instances*.

| Method | MMLU | MATH |
|---|---|---|
| Aggregation-Only | 64.7 | 44.0 |
| Refinement-Only | 60.9 | 38.8 |
| MAGICORE | **67.3** | **46.0** |

*all* instances, regardless of the problem difficulty. We find that uniformly applying refinement actually degrades performance; comparing Weighted SC (the "Aggregation-Only" in row 1) to refinement-only (row 2), we see that refining all samples leads to $3.8\%$ and $5.2\%$ drops on MMLU and MATH, respectively, pointing to the over-correction issue. Conversely, one iteration of our selective refinement (row 3) targets only the challenging instances where the weighted majority vote is unlikely to succeed, resulting in up to $2.6\%$ improvement compared to uniformly applying aggregation (row 1). This demonstrates that our selective refinement not only avoids over-correction but also enhances overall performance by effectively allocating more resources to harder problems.

**PRM-based targeted feedback enables better refinement.** Having demonstrated that selectively applying refinement in MAGICORE is crucial for achieving improvements, we now compare the refinement process with and without using a PRM. To this end, *without using a PRM*, we ask the LLM to generate an updated solution based on its own previous reasoning, referring to this as LLM Self-Refinement. This is similar to SR + SC in Table 1, except we use MAGICORE's ability to identify hard instances and only apply refinement to

Table 3: Comparison of different refinement variants in MAGICORE. Using PRM scores for refinement performs the best.

| Refinement Variants | MMLU | MATH |
|---|---|---|
| LLM Self-Refinement | 65.9 | 44.4 |
| Random Step Score | 66.4 | 43.8 |
| ORM Score (No Step Score) | 66.8 | 45.2 |
| MAGICORE (PRM Step Score) | **67.3** | **46.0** |

those, resorting to Weighted SC for the easier problems. Compared to the full MAGICORE in row 4 of Table 3, using LLM's self-refinement (row 1) results in an average drop of $1.5\%$, indicating that using the LLM for refinement is less effective than using a PRM. To further examine how sensitive the refinement process is to the score quality, in row 2, we replace the actual PRM scores with random scores. The result is worse than row 4, indicating that PRM scores help in localizing errors. Finally, we test whether the global ORM score can offer a similar advantage as using the local PRM score. In row 3, instead of appending the step-wise score in each step, we place the global ORM score at the end of the response to generate the feedback. Result shows that it performs slightly worse than using

the PRM score, suggesting that while global correctness is a strong signal as well, local correctness scores help the LLM to identify and correct errors more effectively. Overall, MAGICORE's use of the step-wise feedback from the PRM is most effective for refinement.

**MAGICORE's two conditions for classifying problem difficulty is more effective than prompting LLMs or any condition alone.** In MAGICORE, we use reward models to classify each instance as easy or hard. Given that the RMs are also fine-tuned LLMs, we investigate whether prompting the LLM to perform this classification directly could replace the external RMs. We compare two settings in the first two rows, where we prompt Llama-3-8B to evaluate the difficulty of an instance. In the first setting (classification), the LLM generates a binary label.

Table 4: Comparison of different ways of detecting hard problems.

| Criterion for Refinement | MMLU | MATH |
|---|---|---|
| Prompt (classification) | 65.2 | 45.0 |
| Prompt (confidence) | 64.7 | 44.4 |
| Condition 1 only | 66.4 | 43.6 |
| Condition 2 only | 66.1 | 44.2 |
| Cond. 1 & Cond. 2 | **67.3** | **46.0** |

In the second setting (confidence), it produces a confidence score ranging from 0 to 1, indicating whether refinement is required – that is, whether the example is easy or hard. Results in Table 4 show that the LLM is less effective at determining instance difficulty compared to a reward model, as evidenced by a performance drop of $1.6\% - 2.6\%$. In rows 3 and 4, we also examine the performance when only one of the conditions of MAGICORE (c.f. Section 2.2) is used to decide difficulty. Specifically, when only condition 1 is applied, an instance is classified as hard if the majority answer's quality is low. Conversely, when only condition 2 is applied, an instance is classified as hard if the RM's answer confidence is low, regardless of the majority answer's quality. Results indicate that while each condition individually outperforms LLM self-verification, combining both yields the best performance. Indeed, in Appendix E, we find that MAGICORE's assessment of problem difficulty shows the highest agreement with human-annotated labels.

**All three issues must be addressed jointly.** To investigate the importance of each refinement issue, we conduct an ablation study to address each issue individually in Table 5 under the following settings: (1) *Only Address Issue 1 (Excessive Refinement)*: Here, we apply selective refinement only, without PRM step-wise scores for feedback generation and without iterations. (2) *Only Address Issue 2 (Inability to Localize and Address Errors)*: Here, we use PRM scores for feedback generation and refine all instances uniformly (i.e. no selective refinement) for

Table 5: Ablation study on addressing each refinement (c.f. Fig. 1) issue one at a time.

| Method | MMLU | MATH |
|---|---|---|
| Only Address Issue 1 | 64.7 | 44.0 |
| Only Address Issue 2 | 65.9 | 45.4 |
| Only Address Issue 3 | 60.3 | 36.4 |
| MAgICoRE (Iter = 3) | **68.9** | **47.8** |

one iteration (i.e. the ablation does not address insufficient refinement). (3) *Only Address Issue 3 (Insufficient Refinement)*: Here, we iteratively refine all samples without incorporating PRM scores (i.e. no error localization) and without performing selective refinement. The results show that only addressing one single refinement issue at a time leads to performance drop, thereby highlighting the need for a joint solution (as we proposed in MAGICORE). We find that only addressing insufficient refinement (Issue 3) causes the highest degradation in performance across tasks, as it fails to efficiently localize errors (without the help of PRM) and also performs excessive refinement. This indicates that Issue 1 and Issue 2 may contribute more to performance than Issue 3, since only addressing Issue 3 has the worst performance.

**MAGICORE generalizes to other domains.** Table 1 shows the benefits of MAGICORE on math reasoning; however, LLMs have been applied to a wide variety of tasks beyond math. Here, we explore expanding MAGICORE to other domains, specifically to a commonsense reasoning task: ARC-Challenge (Clark et al., 2018), and a logical reasoning task: Date Understanding (Srivastava et al., 2022). We sample 200 instances from each dataset and use GPT4o-mini as a PRM for the experiments, as existing standalone PRMs generally only exist for math reasoning. Specifically, we prompt GPT4o-mini to provide step-wise correctness scores without any textual explanations or reasoning, acting the same as a

Table 6: MAGICORE also generalizes to commonsense reasoning and logical reasoning tasks.

| Method | ARC | Date |
|---|---|---|
| Zero-shot | 66.5 | 52.5 |
| 40-way SC | 85.5 | 72.5 |
| 120-way SC | 86.0 | 72.5 |
| MAgICoRE (Iter = 1) | 87.5 | 79.5 |
| MAgICoRE (Iter = 2) | 88.0 | 79.5 |
| MAgICoRE (Iter = 3) | **88.5** | **80.5** |

PRM. The prompt is provided in Appendix G. This approach ensures that our agents do not have access to explanations from a stronger model, and we manually verify that explanations are not

included with the scores. Table 6 shows that MAGICORE transfers to commonsense and logical reasoning, outperforming 120-say SC by 2.5% on commonsense reasoning and 8.0% on logic.

**MAGICORE also improves stronger models like GPT4o-mini.** Table 1 shows results with GPT-3.5-Turbo; here, we show that MAGICORE scales to larger and stronger models as well. Specifically, we run MAGICORE using GPT4o-mini on a subset of MATH data. Due to the high cost of running GPT4o-mini with $k = 120$ and the fact that Fig. 4 shows *decreasing* performance at $k = 120$ for MATH, we only compare to the 40-way SC with the weighted variation that incorporates PRM scores for vote weighting (40-way SC + PRM). Table 7 demonstrates that MAGICORE can also enhance stronger model's performance, albeit with a smaller margin of improvement compared to Llama3-8B and GPT-3.5 shown in Table 1.

Table 7: MAGICORE can also improve GPT4o-mini on MATH.

| Method | Accuracy |
| --- | --- |
| Zero-shot | 72.0 |
| 40-way SC | 79.2 |
| 40-way SC + PRM | 79.4 |
| MAgICoRE (Iter = 1) | 80.2 |
| MAgICoRE (Iter = 2) | **80.4** |
| MAgICoRE (Iter = 3) | **80.4** |

**Separating Reviewer and Refiner roles outperforms combining these roles.** In Table 8, we examine the effects of combining the roles of Reviewer and Refiner by merging their prompts, instructing the model to simultaneously generate both feedback and a refined solution. This method is referred to as "Joint Roles". In MAGICORE, the Reviewer and Refiner have distinct, clearly defined roles, which we refer to as the "Distinct Agents"

Table 8: MAGICORE's separation of the Reviewer and Refiner roles is more effective than combining them into a single role.

| | MMLU | MATH |
| --- | --- | --- |
| Joint Roles | 66.7 | 44.8 |
| Distinct Agents (Ours) | **67.3** | **46.0** |

approach. As before, the performance comparison is based on the first iteration, with all other variables held constant. Our findings show that maintaining separate roles (as in our multi-agent setup) leads to better performance, with the "Joint Roles" configuration resulting in a 0.6% drop in MMLU and a 1.2% decrease in MATH. The larger drop in MATH suggests that its problems are more complex and often require extended reasoning, making the combined roles less effective, whereas maintaining separate roles proves to be more beneficial.

**Discussion of external reward models.** External reward models play an important role MAGICORE and are used in the solutions to all three problems (excessive refinement, inability to localize and address errors, and insufficient refinement). While MAGICORE does utilize external reward models, our framework is modular and can readily incorporate new reward models as they emerge. As the community is actively advancing the performance of reward models evidenced by a benchmark for reward models (Lambert et al., 2024), MAGICORE is thus complementary to and enhanced by progress in reward modeling, rather than constrained by it. While it is possible to train a custom error-identification model, this approach is often data-dependent and prone to obsolescence. In contrast, MAGICORE's modular design overcomes this limitation by enabling the integration of new state-of-the-art models as they become available. Moreover, our experiments in Table 6 indicate that when trained reward models are unavailable, we can use sufficiently strong LLMs in place of trained RMs. For example, we use GPT4o-mini as a reward model for commonsense and logical reasoning.

# 5 CONCLUSION

Building on the observation that different problems require varying amounts of computation, we introduced MAGICORE, a method that adaptively allocates more computational resources to more challenging problems and selectively applies refinement where appropriate, i.e., on harder problems. MAGICORE addresses three key issues in refinement: excessive refinement on easy examples, the inability of LLMs to detect and correct errors in their reasoning, and insufficient refinement on hard instances. Our approach tackles these issues by employing both global and local external reward models to decide which samples to refine. We then incorporate local correctness scores to generate targeted feedback and an iterative multi-agent communication framework to refine solutions for hard problems. Our results across five math datasets and two models show that our coarse-to-fine method consistently outperforms both coarse-grained aggregation and fine-grained refinement alone at any given budget and even outperforms baselines using substantially more computation. In our ablations, we demonstrate the importance of selective refinement, showing that performance generally drops when refining all samples uniformly. We also highlight the role of iteration in our framework, showing increased performance across iterations even as baselines stagnate.

ETHICS STATEMENT

In this work, we propose an inference-time method, MAGICORE, which operates without the need for additional training or fine-tuning. Consequently, the LLMs utilized by MAGICORE (e.g., Llama-3, GPT-3.5) may still exhibit stereotypes, biases, and other negative traits inherent in their pre-training data (Weidinger et al., 2021), over which we have no control. Therefore, the outputs produced by MAGICORE carry the same potential for misuse as those from other test-time methods. Further research is necessary to assess and mitigate these biases in LLMs.

REPRODUCIBILITY STATEMENT

We are making our code available in the supplementary materials to enable replication of our findings. We also provide implementation details of MAGICORE in Section 3 and prompt in Appendix G. The datasets we use are all publicly available (MMLU, MATH, SVAMP, GSM8K, SAT, ARC, Date).

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

## A    RELATED WORK

**Improving Reasoning by Aggregation.** Self-Consistency (SC; Wang et al., 2022) generates $k$ reasoning chains and marginalizes over the chains to obtain answer clusters; the most frequent answer is selected as the final prediction. While simple and effective, it generates $k$ solutions *for every sample*, as both past work and our work show that SC saturates when $k$ increases (Chen et al., 2024b; Li et al., 2024a). Several studies adaptively determine the number of samples ($k$) required *for each instance* to address this (Aggarwal et al., 2023; Li et al., 2024b). However, the performance of such approaches is upper-bounded by SC – they address the cost issue but do not enhance overall performance. To surpass SC, Yin et al. (2024) propose using the LLM to evaluate answer clusters, taking into account both frequency and the LLM-evaluated quality of the answers. Instead, we propose using external RMs to decide between coarse-grained aggregation and fine-grained refinement, identify errors, and aid refinement; this allows us to improve over aggregation or refinement alone.

**LLM-based Verification and Refinement.** Past work mostly uses RMs for verification purposes (Li et al., 2023; Khalifa et al., 2023; Cobbe et al., 2021; Lightman et al., 2023). Havrilla et al. (2024) considers local correctness for refinement in a non-adaptive way and requires specific data curation with fine-tuning, while our work is adaptive and uses off-the-shelf global and local models. Another line of work has proposed using the LLM itself as a verifier, in place of an RM (Liu et al., 2024; Zhang et al., 2024; Aggarwal et al., 2023; Madaan et al., 2023). However, recent work shows the inability of LLMs to "self-verify" their own reasoning (Huang et al., 2024; Stechly et al., 2023; Kamoi et al., 2024; Tyen et al., 2024; Kamoi et al., 2024). Hence, MAGICORE uses external global and local reward models (Wang et al., 2023; Cai et al., 2024) for selective coarse-to-fine refinement. Shridhar et al. (2024) trained specific models to decide when to refine and when to trust refined solutions. This contrasts with our method, where the decision to refine is based on a coarse-to-fine resource allocation method that differentiates easy from hard problems (for which we use global and local reward models), and where refinement is done based on off-the-shelf models. We also iteratively refine guided by global correctness to ensure sufficient refinement, whereas Shridhar et al. (2024) refine only once. Wadhwa et al. (2024) propose a multi-agent detect-critique-refine pipeline for generation tasks. MAGICORE instead focuses on reasoning tasks and uses external RMs for selective coarse-to-fine refinement (whereas Wadhwa et al. (2024)'s detection uses the same metric as their evaluation, which is infeasible in reasoning where the metric – accuracy – requires access to the gold answer).

**Multi-Agent Systems with LLMs.** LLMs can be used in multi-agent systems, where the agents interact, collaborate, and compete (Wang et al., 2024; Lu et al., 2024; Feng et al., 2024). Related to our work, one line of multi-agent research focuses on structured debates or discussions between LLM agents, where the interaction helps refine and improve previously generated solutions (Du et al., 2023; Liang et al., 2023; Yin et al., 2023; Chen et al., 2024a). These studies show improvements over single-agent systems, but a major challenge in multi-agent systems is achieving a correct consensus among LLMs; external feedback can help prevent this consensus from aligning with the agents' internal and possibly erroneous outputs. Therefore, MAGICORE's multi-agent refinement incorporates external RMs for more objective scoring, enabling the generation of targeted feedback for better refinement.

## B    SELF-CONSISTENCY AND WEIGHTED SELF-CONSISTENCY.

Self-Consistency (Wang et al., 2022) is a popular decoding method that uses majority voting to aggregate predictions from different reasoning chains, thus marginalizing over chains. It generates $k$ solutions per question and selects the most frequent final answer from these samples. While simple and effective, this method assigns uniform weight to each reasoning chain, which fails to account for the quality of each solution. To address this limitation, Li et al. (2023) propose Weighted Self-Consistency, accounting for each solution's quality. Formally, both Self-Consistency and Weighted Self-Consistency choose a final answer via:

$$\hat{y} = \arg\max_y \sum_{i=1}^{k} \mathbb{1}_{y_i = y} \mathcal{V}(q; r_i)$$

where $\mathcal{V}(\cdot)$ is a constant 1 in Self-Consistency and the quality measurement (e.g., RM score) in Weighted Self-Consistency. In MAGICORE's final answer selection, we use the sum of the solution-level scores generated by both ORM and PRM as $\mathcal{V}(\cdot)$. Throughout MAGICORE, in cases where we need solution-level PRM scores (compatible with ORM scores), we accumulate the PRM step scores by taking their product (Sun et al., 2024), so that the aggregated PRM score corresponds to a solution.

# C   DETAILS OF THE CONDITIONS

**Condition 1: Is the Majority Answer of High Quality?**   Given a problem $q$, to determine the difficulty of the problem at hand, the Solver generates $k$ solutions $R = \{r_1, \ldots, r_k\}$ and final answers $A = \{a_1, \ldots, a_k\}$ per question and cluster the solutions by their final answer. This produces a partition $\mathcal{A}$ with elements $\mathcal{A}_i$, where $\mathcal{A}_i = \{r_j \in R \mid a_j = a_i\}$. The majority cluster $\mathcal{A}_g$ has the most "votes", i.e., $\mathcal{A}_g = \text{argmax}_{i \in |\mathcal{A}|}|\mathcal{A}_i|$. We evaluate the majority answer quality by both ORM and PRM separately but with the same procedure, as described below. First, we score every reasoning chain $r_i$ within the majority cluster $\mathcal{A}_g$. Both ORM and PRM are able to produce a solution-level score, which we denoted as $\mathcal{S}_i^{\text{RM}}$. Note that we perform this check using ORM and PRM separately, but for simplicity, we use the same notation for solution-level score, which either comes from the ORM or the PRM. We calculate the average of the solution-level scores from the majority group:

$$\mathcal{S}_{avg}^{\text{RM}} = \frac{1}{|\mathcal{A}_g|} \sum_{i=1}^{|\mathcal{A}_g|} \mathcal{S}_i^{\text{RM}}$$

This average score informs us of the majority answer's quality. To set a threshold, we normalize $\mathcal{S}_{avg}^{\text{RM}}$ by using the sample average RM scores (by computing $\mathcal{S}_i^{\text{RM}}$ for each sample and then take the average of these scores). Importantly, this process does not require any labeled data. After normalization, if the average reward of the majority group $\mathcal{S}_{avg}^{\text{RM}} \geq 0$, indicating that the quality of the majority answer is high, Condition 1 will be *true*. Otherwise, if $\mathcal{S}_{avg}^{\text{RM}} < 0$, Condition 1 will be *false*, suggesting that even the most frequent answer is of poor quality and that the instance might benefit from refinement.

**Condition 2: Are Reward Models' Answer Confidence High?**   Besides the quality of the majority answer, we also consider whether the RMs are confident enough in any single answer among the answer clusters. Again we evaluate both ORM and PRM's answer confidence separately but with the same procedure, as described below. First, the answer distribution is formed by (1) the frequency of each unique answer and (2) the total RM score of each answer cluster. We estimate the RM's confidence according to this distribution. If the distribution is concentrated, meaning that only one answer cluster stands out, the RM's answer confidence is treated as high. Conversely, if the distribution is diffused and the clusters' scores are more uniform, then there is no single answer for which the RM has high confidence, i.e., the RM's confidence is low. This motivates a targeted step-wise refinement process to select a more definite answer. Again we use both ORM and PRM to generate the solution-level score $\mathcal{S}_i^{\text{RM}}$. Given the $k$ reasoning chains generated along with the solution-level score, we compute the RM's answer confidence (denoted as $C$) using the entropy of the answer cluster weighed by the RM scores, passing the result through a sigmoid function to normalize it onto $[0, 1]$. Formally, the calculation of the entropy can be expressed as:

$$H = -\sum_{i=1}^{n} p(\mathcal{A}_i) \log p(\mathcal{A}_i); \quad p(\mathcal{A}_i) = \frac{\sum_{i=1}^{|\mathcal{A}_i|} \mathcal{S}_i^{\text{RM}}}{\sum_{\mathcal{A}_j \in \mathcal{A}} \sum_{k=1}^{|\mathcal{A}_j|} \mathcal{S}_k^{\text{RM}}}$$

where $n$ is the number of unique answers among the $k$ chains, $\mathcal{A}_i$ is the $i$-th answer cluster (a set of reasoning chains leading to the same answer) and $\mathcal{A}$ is the set of all clusters. Each answer in a cluster is weighed by its unnormalized solution-level score $\mathcal{S}_i^{\text{RM}}$. To normalize entropy onto a confidence scale, we invert it so that high entropy corresponds to low confidence. We then apply a sigmoid function $\sigma(.)$, mapping the values to the range $[0, 1]$: $C = \sigma(\alpha * (1 - H))$. We set $\alpha$ to 2 to let the distribution stretch more evenly between 0 and 1. This transformation establishes 0.5 as a natural threshold for differentiating low and high confidence, thereby eliminating the need for any threshold tuning. That is, if an instance has $C \geq 0.5$, Condition 2 is *true*, meaning that the RMs are confident on a single answer cluster. Otherwise, if $C < 0.5$, Condition 2 is *false*, suggesting that the RMs' uncertainty among the $k$ chains is high, necessitating a finer refinement.

| Method | MMLU | MATH | SVAMP | GSM8K | SAT | Avg. |
|---|---|---|---|---|---|---|
| 120-way SC | 63.0 | 40.6 | 89.8 | 90.3 | 70.5 | 70.8 |
| 120-way SC + PRM Li et al. (2023) | 65.4 | 44.6 | 90.8 | 90.7 | 72.5 | 72.8 |
| Self-correct + 120-way SC (Kim et al., 2024) | 62.1 | 38.6 | 86.2 | 88.1 | 65.6 | 68.1 |
| Least-to-Most + 120-way SC (Zhou et al., 2023) | 62.6 | 40.6 | 89.0 | 90.3 | 68.9 | 70.3 |
| Multi-Agent Debate + SC (Du et al., 2023) | 64.6 | 41.0 | 89.6 | 90.8 | 72.5 | 71.7 |
| MAgICoRE (Iter=1) | 67.3 | 46.0 | **91.4** | 91.1 | 75.0 | 74.2 |
| MAgICoRE (Iter=2) | 68.4 | 47.2 | 91.1 | **92.3** | 76.4 | 75.1 |
| MAgICoRE (Iter=3) | **68.9** | **47.8** | 91.3 | 91.6 | **78.2** | **75.6** |

Table 9: Performance comparison with additional baselines using `Llama3-8B-Instruct`. Notably, MAGICORE with only one iteration outperforms all baselines despite using fewer samples.

# D    ADDITIONAL EXPERIMENTAL RESULTS

**Comparison with additional baselines.** In addition to Table 1, we also compare with the following baselines: (1) 120-way SC + PRM: The product of step-wise PRM scores is used as the solution-level score. This score is then employed for weighted Self-Consistency, following Li et al. (2023). (2) Self-correct + 120-way SC: We use the "Self-Correct RCI" prompt from Kim et al. (2024) to generate 120 solutions per question, which are subsequently aggregated using Self-Consistency. (3) Least-to-Most + 120-way SC: We use the zero-shot Least-to-Most prompt from Zhou et al. (2023) to generate 120 solutions per question, followed by aggregation via Self-Consistency. (4) Multi-Agent Debate + SC: Following Du et al. (2023), we conduct a three-agent debate over four rounds, repeating this process ten times. The final answers from these ten debates are aggregated using Self-Consistency, yielding 120 generations per question.

We use `Llama3-8B-Instruct` as the base model. Results show that a single iteration of MAGICORE already outperforms methods that rely on PRM for aggregation (120-way SC + PRM), as well as approaches like Self-Correction, advanced prompting, and multi-agent debate. On average, MAGICORE outperforms 120-way SC + PRM by $2.8\%$ despite using fewer samples, highlighting the limitations of using PRM solely for aggregation. Additionally, MAGICORE exceeds Least-to-Most by $5.3\%$, showcasing superior adaptability to problem difficulty. Finally, MAGICORE surpasses Multi-agent Debate by $3.9\%$, indicating that our aggregation and refinement mechanisms scale more effectively at test time.

**Ablations on reward models for final answer selection.** We report MAGICORE up to three iterations in Table 1 and only report the best-performing iteration of Self-Refine + $k$-way SC. Here, we provide extended results in table Table 12. We also conducted another ablation study to evaluate the performance when using ORM, PRM, or a the summation of both scores for final answer selection. As shown in Table 10, utilizing ORM's global correctness score yields better results than aggregating PRM's local correctness score. However, the best performance is achieved when both scores are combined for the final answer aggregation.

Table 10: Ablation study on the final answer selection, using ORM-only, PRM-only or both.

| Aggregation | MMLU | MATH |
|---|---|---|
| ORM-Only | 66.9 | 45.4 |
| PRM-Only | 66.1 | 45.0 |
| Both | **67.3** | **46.0** |

**Reliable step-wise scores enables LLM refinement.** To compare with an oracle PRM, we sample 500 instances from the Math-Shepherd dataset (Wang et al., 2023), which includes gold label correctness for each step. Besides the three settings we evaluated in Table 3, we also evaluate the oracle PRM score, where feedback uses the gold correctness labels. Table 11 shows that the oracle PRM score performs the best, followed by the predicted PRM score, suggesting that given reliable stepwise scores, LLMs can effectively refine their solutions and improve.

Table 11: Comparison of different refinement variants in MAGICORE.

| Refinement Variants | Accuracy |
|---|---|
| No feedback (LLM self-refine) | 48.30 |
| Random PRM score | 49.60 |
| PRM predicted score | 51.20 |
| Oracle PRM score | **52.40** |

|  | MMLU | MATH | SVAMP | GSM8K | SAT | Avg. |
|---|---|---|---|---|---|---|
| **Llama3-8B-Instruct** | | | | | | |
| Zero-shot CoT | 50.4 | 24.2 | 72.4 | 80.1 | 58.2 | 57.1 |
| Self-Refine (Iter=1) | 49.6 | 24.6 | 72.0 | 79.0 | 57.7 | 56.3 |
| Self-Refine (Iter=2) | 50.2 | 23.8 | 72.8 | 79.6 | 59.3 | 57.1 |
| Self-Refine (Iter=3) | 49.8 | 24.0 | 72.6 | 79.6 | 59.6 | 57.1 |
| Best-of-$k$ ($k = 120$) | 62.6 | 41.4 | 88.7 | 90.1 | 72.4 | 71.0 |
| $k$-way SC ($k = 120$) | 63.0 | 40.6 | 89.8 | 90.3 | 70.5 | 70.8 |
| Self-Refine + $k$-way SC (Iter=0) | 62.1 | 40.4 | 88.6 | 90.1 | 68.2 | 69.9 |
| Self-Refine + $k$-way SC (Iter=1) | 61.3 | 40.6 | 88.9 | 89.7 | 67.7 | 69.6 |
| Self-Refine + $k$-way SC (Iter=2) | 62.7 | 40.0 | 88.9 | 90.1 | 68.6 | 70.1 |
| Self-Refine + $k$-way SC (Iter=3) | 62.3 | 41.0 | 89.2 | 90.3 | 68.0 | 70.2 |
| Self-Refine + $k$-way SC (Iter=4) | 62.1 | 41.4 | 89.2 | 90.1 | 67.7 | 70.1 |
| Self-Refine + $k$-way SC (Iter=5) | 62.7 | 40.4 | 88.6 | 89.7 | 67.7 | 69.8 |
| MAGICORE (Iter=1) | 67.3 | 46.0 | **91.4** | 91.1 | 75.0 | 74.2 |
| MAGICORE (Iter=2) | 68.4 | 47.2 | 91.1 | **92.3** | 76.4 | 75.1 |
| MAGICORE (Iter=3) | **68.9** | 47.8 | 91.3 | 91.6 | **78.2** | **75.6** |
| MAGICORE (Iter=4) | **68.9** | **48.0** | 91.3 | 91.1 | **78.2** | 75.5 |
| MAGICORE (Iter=5) | 68.4 | **48.0** | 91.1 | 91.6 | **78.2** | 75.5 |
| **GPT-3.5-Turbo** | | | | | | |
| Zero-shot CoT | 62.5 | 37.2 | 78.1 | 78.5 | 76.8 | 66.6 |
| Self-Refine (Iter=1) | 62.4 | 37.4 | 77.7 | 77.4 | 77.3 | 66.4 |
| Self-Refine (Iter=2) | 61.6 | 37.6 | 78.6 | 77.9 | 76.9 | 66.5 |
| Self-Refine (Iter=3) | 61.1 | 37.4 | 77.9 | 78.4 | 77.1 | 66.4 |
| Best-of-$k$ ($k = 120$) | 70.1 | 50.6 | 87.7 | 90.5 | 87.8 | 77.3 |
| $k$-way SC ($k = 120$) | 70.4 | 51.2 | 86.9 | 89.8 | 87.6 | 77.1 |
| Self-Refine + $k$-way SC (Iter=0) | 69.4 | 49.8 | 86.9 | 88.1 | 85.6 | 76.0 |
| Self-Refine + $k$-way SC (Iter=1) | 69.8 | 49.0 | 87.1 | 88.3 | 85.0 | 75.8 |
| Self-Refine + $k$-way SC (Iter=2) | 70.1 | 49.4 | 88.1 | 88.1 | 84.5 | 76.0 |
| Self-Refine + $k$-way SC (Iter=3) | 69.6 | 48.8 | 87.3 | 87.8 | 85.2 | 75.7 |
| Self-Refine + $k$-way SC (Iter=4) | 69.8 | 48.4 | 87.1 | 87.1 | 85.0 | 75.5 |
| Self-Refine + $k$-way SC (Iter=5) | 69.6 | 48.6 | 87.3 | 87.4 | 84.5 | 75.5 |
| MAGICORE (Iter=1) | **73.7** | 57.2 | 89.4 | 91.1 | 90.1 | 80.3 |
| MAGICORE (Iter=2) | 73.3 | 57.8 | **90.1** | 91.1 | **90.9** | 80.6 |
| MAGICORE (Iter=3) | 73.6 | **58.6** | **90.1** | **91.4** | **90.9** | **80.9** |
| MAGICORE (Iter=4) | 73.6 | 58.0 | 89.9 | **91.4** | **90.9** | 80.8 |
| MAGICORE (Iter=5) | 73.4 | 57.6 | 89.4 | 91.1 | **90.9** | 80.5 |

Table 12: Extended version of Table 1. Here we show all more iterations for Self-Refine + $k$-way SC and MAGICORE. While SR + SC does not show a clear improvement with more iterations, MAGICORE continues to improve, peaking at the third iteration.

# E    MODEL-PREDICTED VS. HUMAN-ANNOTATED PROBLEM DIFFICULTY

We analyze the model's prediction of problem difficulty. Specifically, we utilize the MATH dataset, which includes human-annotated difficulty levels ranging from 1 to 5, with higher levels indicating increased problem complexity. For our analysis, we split the problems as follows: (1) Easy: Levels 1 and 2 and (2) Hard: Levels 4 and 5. We exclude Level 3 problems to create a clearer distinction between easy and hard categories. We compare the overlap between our model's predictions and these human-annotated levels. We treat hard as the positive label. The results are presented in Table 13. To provide a comparative analysis, we include: (1) a random baseline that assigns easy and hard labels at random, (2) a prompt-based baseline that directly prompts the LLM to classify the problem difficulty, and (3) another prompt-based baseline that prompts the LLM to generate a confidence score when answering, where a confidence score of $\geq 0.5$ is classified as "easy". Results show that our conditions substantially outperform all baselines. Interestingly, the prompt-based methods perform worse than the random baseline, particularly the one relying on confidence scores, which classifies *all problems as easy*; this method scores 0 for both precision and recall since we treat "hard" as the positive label, so it has 0 true positives. This suggests that our framework is highly effective at distinguishing true problem difficulty based on the conditions outlined in our methodology.

|  | P | R | F1 |
|---|---|---|---|
| Random | 68.4 | 49.6 | 57.5 |
| Prompt-based (classification) | 65.9 | 10.3 | 17.8 |
| Prompt-based (confidence) | 0.0 | 0.0 | 0.0 |
| MAGICORE | **86.3** | **67.6** | **75.8** |

Table 13: The Precision (P), Recall (R) and F1 of the model predicted problem difficulty.

# F    TOKEN COUNT ANALYSIS

In Fig. 4, we are mainly comparing the number of generations ($k$) per question with the baselines. To provide a more granular analysis, we break down the generations at the token level and compare costs in terms of token counts. The results are detailed in Fig. 5. For Self-Consistency, the input tokens are counted only once per question, as it uses the same input to generate $k$ responses. In contrast, the input token count for MAGICORE includes all prompts across all agents – Solver, Reviewer, and Refiner. We also include the token count for the ORM and PRM in MAGICORE. Since the cost of input tokens is typically $0.25\times$ that of output tokens[4], we present the normalized total token cost as $0.25\times$ input $+ 1\times$ output. Results in Fig. 5 show that (1) scaling Self-Consistency from $k = 40$ to $k = 120$ largely increases token overhead while yielding marginal improvements. (2) MAGICORE exhibits superior scalability, achieving substantially higher performance gains with increased token usage. On MMLU, MATH and SAT, we observe a clear upward trend with an increased token count; MAGICORE consistently improves with additional tokens (unlike SC which tends to stagnate). (3) The first iteration of MAGICORE outperforms 120-way SC fewer tokens.

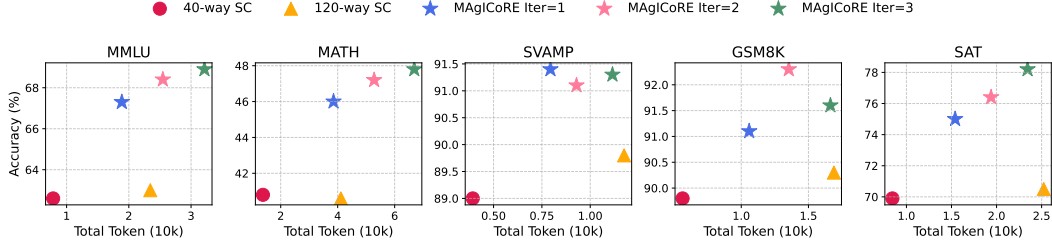

Figure 5: Token count comparison with Self-Consistency across different datasets. Scaling Self-Consistency from $k = 40$ to $k = 120$ introduces substantial token overhead while providing marginal improvements. In contrast, MAGICORE demonstrates superior scalability, delivering much higher performance gains with an increased token count. Notably, the first iteration of MAGICORE consistently outperforms 120-way SC while using fewer tokens.

---

[4]See https://openai.com/api/pricing, https://www.anthropic.com/pricing#anthropic-api, and https://ai.google.dev/pricing#1_5pro

## G    PROMPT FOR THE REVIEWER AND THE REFINER

---

**Reviewer's Prompt**

Your task is to provide step-by-step feedback to the current solution.

You will be given a math problem and a current solution, along with the scores for each step based on its correctness.

- You will find (Score: n/10) at the end of each step.

- The maximum (best) score is 10, which means that this step is 100% correct (and 0% incorrect).

- The minimum (worst) score is 0, which means that this step is 100% incorrect (and 0% correct).

- Pay attention to the steps having scores lower than 6, and carefully identify the errors in those steps.

- Provide your explanation of the error and how it can be fixed. DO NOT propose a new solution, just the explanation.

Question:

{question}

Current Solution:

{solution}

Let's review the steps with lower scores and identify the errors.

{feedback}

---

**Refiner's Prompt**

Your task is to fix the error in the given solution, based on the teacher's feedback.

- After reviewing the solution and feedback, provide a better solution to fix the mistakes you found.

Question:

{question}

Current Solution:

{solution}

Teacher Feedback:

{reviewer_feedback}

Revised Solution:

Let's review the solution and the feedback, and then refine the current solution.

After reviewing the solution and feedback, let's solve this math problem again.

{refined_solution}

---

## H  PROMPT FOR GPT4O-MINI AS PRM

> **GPT4o-mini as PRM Prompt**
>
> You will be given a question and its corresponding step-by-step solution.
>
> Your task is to evaluate the correctness of each step in the solution by assigning a score between 1 and 10.
>
> Your response should be formatted as follows:
>
> score: <comma separated scores>
>
> where the length of the list corresponds to the number of steps in the solution.
>
> Each step is clearly separated in the solution by <Step 1:>, <Step 2:> ... <Step k:>, your output score list should be the exact same length as k.
>
> Question: {question}
>
> Solution: {solution}
>
> score:

## I  QUALITATIVE EXAMPLES OF REFINEMENT

---

QUESTION: Josh decides to try flipping a house. He buys a house for $80,000 and then puts in $50,000 in repairs. This increased the value of the house by 150%. How much profit did he make? (Gold answer: **70000**)

SOLVER'S OUTPUT (BEFORE REFINEMENT):
Step 1: Calculate the value of the house after repairs. The house is worth 80,000 after repairs. It is 150% higher than before. So the value is (1 + 150/100) * 80,000. 80,000 * (3/2) is 120,000. So the house is worth 120,000 after repairs. (Score: 2/10)
Step 2: Calculate the cost of repairs. The cost of repairs is 50,000. (Score: 2/10)
Step 3: Calculate the profit. The profit is the difference between the value after repairs and the cost before repairs. 120,000 - 80,000 - 50,000 is 30,000 (incorrect).

REVIEWER'S OUTPUT:
Step 1: The low score of 2 suggests an error in this step. The problem states that the value of the house increased by 150%, not that the value is 150% higher than before.
Step 2: The cost of repairs is indeed $50,000.
Step 3: The low score of 3 suggests an error in this step. The profit would be: 200,000 (value after repairs) - 80,000 (cost before repairs) - 50,000 (cost of repairs) = 70,000

REFINER'S OUTPUT:
Step 1: Calculate the value of the house after repairs. The house is worth 80,000 after repairs. It is increased by 150%. So the value is 80,000 x (1 + 150/100) = 80,000 x 2.5 = 200,000.
Step 2: Calculate the cost of repairs. The cost of repairs is 50,000.
Step 3: Calculate the profit. The profit is the difference between the value after repairs and the cost before repairs. 200,000 - 80,000 - 50,000 is 70,000 (correct).

---

Table 14: A qualitative example of our multi-agent refinement on GSM8K using `Llama3-8B-Instruct`. The score at the end of each step is provided by the PRM.

QUESTION: The Greek army contained two types of soldiers: the upper class and the lower class soldiers. If there were a total of 5 upper class soldiers, and 10 lower class soldiers in a certain part of Athens, and the battle of Thermopylae demands a force of 4 upper class soldiers and 8 lower class soldiers, how many different battalions can be sent? (Gold answer: **225**)

SOLVER'S OUTPUT (BEFORE REFINEMENT):
Step 1: Find out the number of upper class soldiers left. We have 5 upper class soldiers and we need 4. So we have 5 - 4 = 1 upper class soldiers left. (Score: 9/10)
Step 2: Find out the number of lower class soldiers left. We have 10 lower class soldiers and we need 8. So we have 10 - 8 = 2 lower class soldiers left. (Score: 9/10)
Step 3: Find out the number of different battalions. We have 1 upper class soldier and 2 lower class soldiers. The number of different battalions is the number of ways we can arrange 1 upper class soldier and 2 lower class soldiers. This is 3. (Score: 5/10) (incorrect).

REVIEWER'S OUTPUT:
Step 1 and Step 2: The scores of 9/10 suggest that these steps are mostly correct. However, we should be careful when calculating the number of upper and lower class soldiers left.
Step 3: The low score of 5 suggests an error in this step. The problem asks for the number of different battalions, not the number of ways to arrange the soldiers. The correct solution is to find the number of ways to choose 4 upper class soldiers from the available 5 and 8 lower class soldiers from the available 10.

REFINER'S OUTPUT:
Step 1: Find out the number of upper class soldiers needed. We need 4 upper class soldiers.
Step 2: Find out the number of lower class soldiers needed. We need 8 lower class soldiers.
Step 3: Find out the number of different battalions. We can choose 4 upper class soldiers from the 5 available in 5C4 = 5 ways. We can choose 8 lower class soldiers from the 10 available in 10C8 = 45 ways. The number of different battalions is the number of ways to choose 4 upper class soldiers and 8 lower class soldiers, which is $5 \times 45 = 225$. (correct).

Table 15: A qualitative example of our multi-agent refinement on MATH using `Llama3-8B-Instruct`. The score at the end of each step is provided by the PRM.

# J  QUALITATIVE EXAMPLES OF OVER-CORRECTION

QUESTION: A new program had 60 downloads in the first month. The number of downloads in the second month was three times as many as the downloads in the first month, but then reduced by 30% in the third month. How many downloads did the program have total over the three months? (Gold answer: **366**)

SOLVER'S OUTPUT (BEFORE REFINEMENT):
Step 1: Determine the number of downloads for the second month. The number of downloads for the second month was three times as many as the first month, so it was 60 * 3 = 180. (Score: 10/10)
Step 2: Determine the number of downloads for the third month. There was a reduction of 30%, so we calculate 180 * (1 - 0.30) = 180 * 0.7 = 126. (Score: 10/10)
Step 3: Determine the total number of downloads for the three months. There were 60 downloads in the first month, and 180 downloads in the second month. So the total number of downloads for the first two months is 60 + 180 = 240. Then, there were 126 downloads in the third month. So the total number of downloads over the three months is 240 + 126 = 366. (Score: 6/10) (correct).

REVIEWER'S OUTPUT:
Step 1-2: The high scores of 10 suggest that these steps are correct.
Step 3: The score of 6 suggests that this step is mostly correct. However, the improvement you could make is to explicitly state the calculation for the total number of downloads, rather than just stating the result.

REFINER'S OUTPUT:
Step 1: Determine the downloads in the second month. Three times 60 is 3 x 60 = 180. Then 30% of 180 is 180 x 0.3 = 54. 180 - 54 is 126. So 126 downloads in the second month.
Step 2: Determine the total downloads over the three months. 60 downloads in the first month, and 126 downloads in the second month, and 126 - 54 = 72 downloads in the third month. 60 + 126 + 72 is 258. The answer is 258. (incorrect).

Table 16: A qualitative example of over-correction on GSM8K using `Llama3-8B-Instruct`.

QUESTION: If $f(x) = \frac{3x-2}{x-2}$, what is the value of $f(-2) + f(-1) + f(0)$? Express your answer as a common fraction. (Gold answer: $\frac{14}{3}$)

SOLVER'S OUTPUT (BEFORE REFINEMENT):
Step 1: Determine the values of $f(-2)$, $f(-1)$, and $f(0)$ by plugging the values into the expression for $f(x)$.
$f(-2) = \frac{3(-2)-2}{-2-2} = \frac{-6-2}{-4} = \frac{-8}{-4} = 2,$

$f(-1) = \frac{3(-1)-2}{-1-2} = \frac{-3-2}{-3} = \frac{-5}{-3} = \frac{5}{3},$

$f(0) = \frac{3(0)-2}{0-2} = \frac{0-2}{-2} = \frac{-2}{-2} = 1.$ (Score: 10/10)
Step 2: Add the values of $f(-2)$, $f(-1)$, and $f(0)$ to obtain $f(-2) + f(-1) + f(0) = 2 + \frac{5}{3} + 1$

$\frac{6}{3} + \frac{5}{3} + \frac{3}{3} = \frac{14}{3}.$

Final Answer: The final answer is $\boxed{\dfrac{14}{3}}$. I hope it is correct. (Score: 7/10) (correct).

REVIEWER'S OUTPUT:
Step 1: The scores of 10/10 suggest that this step is correct.
Step 2: The high score of 7 suggests that this step is mostly correct. However, it might be beneficial to include intermediate steps for clarity.

REFINER'S OUTPUT:
Step 1: We can calculate $f(-2)$, $f(-1)$, and $f(0)$ using the given expression for $f(x)$: $f(-2) = \frac{3(-2)-2}{-2-2} = \frac{-4}{-4} = 1,$

$f(-1) = \frac{3(-1)-2}{-1-2} = \frac{-3}{-3} = 1,$

$f(0) = \frac{3(0)-2}{0-2} = \frac{-2}{-2} = 1.$
Step 2: Now we can add these values together: $f(-2) + f(-1) + f(0) = 1 + 1 + 1 = \frac{3}{1}$

Final Answer: The final answer is $\boxed{\dfrac{3}{1}}$. I hope it is correct. (incorrect).

Table 17: A qualitative example of over-correction on MATH using `Llama3-8B-Instruct`.

