# OpenReview forum: "MAgICoRe: Multi-Agent, Iterative, Coarse-to-Fine Refinement for Reasoning"
_ICLR.cc/2025/Conference — Submitted to ICLR 2025_

### Official Review · Reviewer_u3Va · 2024-11-02

**Soundness:** 2
**Presentation:** 2
**Contribution:** 2
**Rating:** 5
**Confidence:** 5

**Summary:**

The authors focus on achieving reliable problem-solving with LLMs through refinement. They identify three main issues in existing refinement methods: excessive refinement, inability to localize and address errors, and insufficient refinement. To address these challenges, they propose MAGICORE, an adaptive framework designed to enhance both performance and efficiency in multi-step reasoning with LLMs by intelligently applying test-time aggregation and targeted refinement. Experimental results show that MAGICORE outperforms leading alternative approaches by a significant margin.

**Strengths:**

The paper is easy to follow and includes figures explaining the intuitive concept behind the proposed framework.

The authors propose a framework aiming to simultaneously address all issues in the existing refinement approaches. In particular, the authors try to design a complex process to avoid or alleviate the problems related to each issue.

The experimental results presented in the paper demonstrate the state-of-the-art performance of the proposed framework.

**Weaknesses:**

1. The major issue of this paper is the lack of research focus and key insights. At first glance, the authors open a big question aiming to address all issues in the existing refinement approaches. However, this question is too big to hold for both the researcher and the reader because understanding and gaining insights into one issue to capture solid motivations may be challenging enough. For instance, in the introduction, even Figure 1 contains too much content to understand, and it is hard for me to figure out which major problem (based on your research experience) should be the most critical. Interestingly, in this submission, the authors emphasize the importance of identifying difficult problems and allocating more resources to them (i.e., 'over-correction'). However, in this paper, they present everything uniformly, making each part overly complex in both conceptual explanation and writing.

2. The consequence of presenting everything together is that, to me, the paper's main contribution to the community becomes difficult to identify. It is difficult to discern the in-depth insights the authors gained from related work that motivated their approach design. Specifically, can the authors claim that the proposed framework addresses all three issues present in existing approaches? If not, which issue does the framework address more effectively, and which one remains less resolved? If the framework can address one issue well, why and what makes this framework achieve better performance in this issue?

3. Besides, I may not fully capture the relation or connections between each module of this framework. By saying connection, I mean that why the next module is necessary after the previous one finishes. For example, after identifying hard problems based on the RM scores, why do we need to perform a complex iteration multi-agent process? Is it good enough for me to use an existing complex approach to address these hard problems? I believe this misunderstanding stems from the complexity of the framework and the authors' vague problem definition. In most moments during the paper reading, before I fully appreciate and explore a module in depth, the author has already started defining a new problem and proposing a more complex solution.

4. More importantly, with so many equally important agents involved, hallucinations could become a significant issue: if any agent produces an error, it may impact other modules. Worse, based on my experience, the issues caused by hallucinations tend to accumulate over iterations rather than being resolved. However, the authors do not discuss this obvious core drawback of LLMs. For example, some existing work suggests that their framework relies on advanced LLMs and performs poorly with weaker LLMs. The authors of this paper fail to provide such a necessary discussion.

5. Additionally, while reading the paper, I noticed that the authors provide only brief explanations without engaging in in-depth discussions before quickly moving on. As a result, I am left wondering what findings or insights, beyond these design elements, might better illuminate the corresponding issues. This is because engaging in in-depth, specific discussions rather than quickly presenting design elements should be a core focus within the community.


6. In aiming to address all three issues by incorporating these modules into the framework, the current experiments are insufficient to demonstrate its effectiveness or provide a deeper understanding. For instance, we might ask about the 'cost' of this framework. By 'cost,' I refer to the number of interactions required and the number or length of prompts needed to solve the problem. Second, a more specific ablation study may be necessary to examine the three issues in existing approaches, identifying which issue is most severe and to what extent each part of the framework addresses it. Besides, if 1-2 issues are ignored, I can remove or simplify the corresponding modules from the framework. Then, how about the performance? Third, please include more state-of-the-art approaches in the baseline, as Self-Refine may be outdated given the recent advancements in the field (refinement).


7. Last but not least, I believe this paper does not align with my preferences, as the authors seem to combine multiple novel ideas into a single framework, which can be represented as A + B + C... Due to this, in addition to lacking experience in addressing a major, well-defined problem step-by-step, I feel there are some overclaims in the paper.  --- all three issues mentioned in the paper can be addressed (again without in-depth insights).

**Questions:**

See weaknesses above.

---

> ### Author Response · Authors · 2024-11-20
> **Response to Reviewer u3Va (Part 1)**
>
> We appreciate the reviewer's thorough feedback. To answer the question regarding research focus, whether the proposed framework addresses all three issues, and what are the connections between our proposed modules, we would like to highlight that refinement is a multi-faceted process with several challenges. For example, applying uniform refinement across all samples risks over-correction, which can degrade overall performance. At the same time, refining “the right samples” only once may be insufficient. As a result, a framework that aims to address excessive refinement may end up suffering from under-refinement. Besides this tradeoff and beyond identifying *which* samples require refinement, another critical question is *how* to refine them effectively. MAgICoRE jointly tackles these challenges with its multi-agent design and the incorporation of reward models.
>
> To further illustrate this, **we conduct an additional ablation to illustrate the importance of tackling these issues jointly**. In the ablation, we only address one issue at a time. The following table refers to the settings:
>
> |                     Method                   | Selective Refinement | PRM Fine-Grained Feedback | Iterative Refinement |
> |-----------------------------------------------------|:--------------------:|:-------------------------:|:--------------------:|
> | Only Address Issue 1  (Excessive Refinement)        |           yes          |             no             |           no          |
> | Only Address Issue 2  (Inability to Localize Error) |           no          |             yes             |           no          |
> | Only Address Issue 3 (Insufficient Refinement)      |           no          |             no             |           yes          |
>
> And the following are the results.
>
> | Method | MMLU | MATH |
> |---------|--------|------|
> | Only address issue 1 | 64.7 | 44.0 |
> | Only address issue 2 | 65.9 | 45.4 |
> | Only address issue 3 | 60.3 | 36.4 |
> | MAgICoRE (Iter=3)    | **68.9** | **47.8** |
>
> The results show that
> - “Only Address Issue 1” is less effective due to the absence of PRM-based targeted feedback and iterative refinement.
> - “Only Address Issue 2” incorporates PRM scores for all samples, but this sacrifices efficiency and still underperforms the full setup used in MAgICoRE.
> - “Only Address Issue 3” performs the worst, as it uniformly refines all samples without leveraging PRM-based targeted feedback.
> Thus, all three issues must be addressed jointly, as MAGiCoRE does. These results are further described in lines 455-472 in the revised pdf.
>
> > More in-depth discussions should be placed in the paper
>
> We appreciate the reviewer’s valuable feedback on the writing, and we made several improvements to enhance clarity: (1) moved related work to the appendix to relax the space limit. (2) added more method details and motivations to Section 3.2. (3) stressed that the three refinement issues are tied and that we are jointly addressing the three issues instead of independently. These modifications are highlighted in blue in the revised version.
>
> > Why do we need to perform a complex iteration multi-agent process?
>
> The main idea of the iterative multi-agent framework is to break the refinement process into two stages: feedback generation and refinement, so that the feedback can incorporate PRM step-wise scores to enable better error localization. Empirically, our results in Table 1 and Figure 3 in the original version demonstrate that the proposed multi-agent refinement approach performs better and continues to improve with additional iterations. In contrast, Figure 3 shows that baseline methods like Self-Consistency and Best-of-k fail to benefit from the iterative process. While any test-time scaling method can, in theory, be applied to this subset of hard samples based on our classification, our findings indicate that simply increasing the number of responses per question (as seen in the 40-way and 120-way Self-Consistency experiments), yields limited benefits. On the other hand, the proposed multi-agent framework is more effective.

---

> > ### Author Response · Authors · 2024-11-20
> > **Response to Reviewer u3Va (Part 2)**
> >
> > > The issues caused by hallucinations tend to accumulate over iterations rather than being resolved.
> >
> > **We manually reviewed 20 pairs of feedback and refined solutions, and only found 1 instance containing hallucination.** Here we refer to [1] who only count “statement not mentioned in the problem” as a hallucination, excluding calculation errors.
> >
> > In addition to finding a low percentage (1/20) of hallucination, we note that in each refinement iteration, we have an Outcome Reward Model (ORM) and a Process Reward Model (PRM), each serving distinct purposes. The PRM provides step-wise scores that guide the Reviewer agent in generating targeted feedback, while the ORM evaluates whether the refined solution has improved by the end of the iteration. Since we keep only the top-k solutions determined by the ORM, it ensures that the other k suboptimal solutions are discarded, rather than expending resources on refining flawed outputs and accumulating potential hallucinations and errors.
> >
> >
> > [1] https://arxiv.org/abs/2212.07919
> >
> > > What is the cost of the framework
> >
> > We include a token count comparison in **Figure 5** of the revised version. We find that scaling 40-way Self-Consistency (SC) to 120-way SC largely increases token usage but yields a limited performance improvement (and even a drop on MATH). In contrast, **MAgICoRE effectively improves the performance with a larger token budget**. Notably, MAgICoRE uses fewer tokens on datasets such as SVAMP, GSM8K, and SAT compared to 120-way SC, while achieving superior performance. We also added these details in lines 995-1021 in the revised pdf.
> >
> > > Please include more state-of-the-art approaches in the baseline
> >
> > We have added the following new baselines for comparison. These include baselines also using a PRM, specifically **LLM self-correction [2], Least-to-Most prompting [3], and multi-agent debate [4]**. None of these baselines outperform MAgICoRE and we describe these results in detail in lines 864-895 and summarize the setup and findings below.
> >
> > 1. **120-way SC + PRM**: In this baseline, the product of step-wise PRM scores is used as the solution-level score. This score is then employed for weighted Self-Consistency, following [5].
> > 2. **Self-correct + 120-way SC**: We use the Self-Correct RCI prompt from [2] to generate 120 solutions per question, which are subsequently aggregated using Self-Consistency.
> > 3. **Least-to-Most + 120-way SC**: This baseline employs the zero-shot Least-to-Most prompt from [3] to generate 120 solutions per question, followed by aggregation via Self-Consistency.
> > 4. **Multi-Agent Debate + SC**: We conduct a 3-agent debate [4] over four rounds, repeating this process ten times. The final answers from these ten debates are aggregated using Self-Consistency, yielding 120 generations per question.
> >
> >
> > | Method | MMLU | MATH | SVAMP | GSM8K | SAT | Avg. |
> > |--------|-------|-------|-------|-------|-------|-------|
> > | 120-way SC | 63.0 | 40.6 | 89.8 | 90.3 | 70.5 | 70.8 |
> > | 120-way SC + PRM [1] | 65.4 | 44.6 | 90.8 | 90.7 | 72.5 | 72.8 |
> > | Self-correct + 120-way SC [2] | 62.1 | 38.6 | 86.2 | 88.1 | 65.6 | 68.1 |
> > | Least-to-Most + 120-way SC [3] | 62.6 | 40.6 | 89.0 | 90.3 | 68.9 | 70.3 |
> > | Multi-Agent Debate + SC [4] | 64.6 | 41.0 | 89.6 | 90.8 | 72.5 | 71.7 |
> > | MAGICORE (Iter=1) | 67.3 | 46.0 | **91.4** | 91.1 | 75.0 | 74.2 |
> > | MAGICORE (Iter=2) | 68.4 | 47.2 | 91.1 | **92.3** | 76.4 | 75.1 |
> > | MAGICORE (Iter=3) | **68.9** | **47.8** | 91.3 | 91.6 | **78.2** | **75.6** |
> >
> > [2] https://arxiv.org/abs/2303.17491
> >
> > [3] https://arxiv.org/abs/2205.10625
> >
> > [4] https://arxiv.org/abs/2305.14325
> >
> > [5] https://aclanthology.org/2023.acl-long.291/

---

> > > ### Author Response · Authors · 2024-11-23
> > > **Reminder to reviewer u3Va**
> > >
> > > Thank you once again for your valuable feedback. We hope our response has addressed all of your questions and will allow you to revisit your score, otherwise we would be happy to engage further and address any further questions you might have in the remaining few days of the discussion period.

---

> > > > ### Author Response · Authors · 2024-11-25
> > > > **Reminder to reviewer u3Va**
> > > >
> > > > Since the rebuttal period is drawing to a close, with only 1 day left before the 26th, we wanted to check in again and see if our additional positive results and our responses have addressed your comments. Based on your suggestions, we have added new baselines (which we outperform), have added more ablations and a qualitative analysis, and have sought to address your other comments and questions in detail in our rebuttal. If your concerns have been addressed, we would appreciate if you could revisit your score accordingly.

---

> ### Author Response · Authors · 2024-11-27
> **Reminder for Reviewer u3Va**
>
> Thank you once again for your valuable feedback and suggestions. Since the PDF update window closes in 24 hours, we’d greatly appreciate your confirmation on whether our additional positive results—**MAgICoRE's improvement over 4 new baselines and the detailed token cost comparison that shows its cost-effectiveness**—along with our responses, address your comments. If they do, we kindly ask you to consider revisiting your score.

---

> ### Author Response · Authors · 2024-11-30
> **Reminder for Reviewer u3Va**
>
> Dear Reviewer u3Va,
>
> As the deadline for the discussion period approaches, we wanted to follow up and see if our additional experiments and responses have addressed your comments. If they have, we would greatly appreciate it if you could revisit your score accordingly.
>
> Best,
>
> The Authors

---

### Official Review · Reviewer_cViA · 2024-11-03

**Soundness:** 3
**Presentation:** 2
**Contribution:** 3
**Rating:** 5
**Confidence:** 4

**Summary:**

This paper presents MAgICoRe (Multi-Agent, Iterative, Coarse-to-Fine Refinement), a framework designed to enhance the answer quality of LLMs. The framework addresses three major challenges in refining LLM outputs: avoiding excessive refinement (which can lead to over-correction), targeting error localization, and ensuring sufficient refinement. It proposes to use three agents (solver, reviewer, and refiner), and two reward models (PRM for local scores and ORM for global scores) to enhance the base model’s performance. The key idea behind the proposed framework is categorizing problems as easy or hard, solving easy problems with coarse-grained aggregation, and solving hard ones with fine-grained and iterative multi-agent refinement. The authors evaluate MAgICoRe on 5 datasets and 2 models and show that it obtains accuracy higher than weighted self-consistency.

**Strengths:**

The idea of incorporating RM models and selective refinement with LLMs is very interesting.
The results are very promising, and this method seems to be better than the baselines.
The authors open source their code.

**Weaknesses:**

Writing
W1: The writing of Section 2 could be clearer. Some definitions of the conditions (in Appendix B) are very important to understand the method. Please move it up to the main context. This is especially true when you have a complex Figure 2. If there is no space, please move related works to the appendix.

Novelty
W2: While the reviewer appreciated the proposed framework, the classification of this method is not very clear. If we borrow the classification as used by [1], do the authors consider this work as an intrinsic or extrinsic refinement? If intrinsic, then you should not use separate RM models (instead you should finetune Llama3-8b to be your RM). If extrinsic, then this method seems less accurate than, say, RAG-assisted refinement.

W3: The framework is a combination of multi-agent and RM for selective refinement.  The novelty of the entire system may be hard to justify, as individual parts of the framework do not seem very new. For example, multi-agent is a very well-studied topic now, and in terms of selective refinement, [2] uses implicit confidence of LLM to selectively refine the response. [3] is less similar, but it also uses confidence to select which LLM to use for prediction. In the context of LLM, confidence is similar to the explicit RM (although the reviewer agrees that RM is better than confidence). However, as the authors admit in the paper, the idea of using RM is also not novel (l153).

Evaluation
W4: SVAMP is a dataset with 700 training and 300 testing samples. The paper says that the authors have evaluated the 1000 samples of SVAMP (l304), and the results (for example 78.1) do seem like that. This is problematic though. Please report the result on the test set only.

W5: Please specify the subtasks in MMLU as used. The reviewer failed to add up the numbers to 974 for the standard math-related questions (l308).

W6: Please add GPT4 for a subset of the test datasets.

W7: The cost of the proposed framework seems very high compared to decoding-only frameworks (majority voting or vanilla self-consistency). The method runs models multiple times, whereas decoding-only frameworks mainly just decode and do not run the main model. Can the authors please provide the inference delay or cost for the comparison?

Reference
[1] Huang, Jie, et al. "Large language models cannot self-correct reasoning yet." arXiv preprint arXiv:2310.01798 (2023).
[2] Li, Loka, et al. "Confidence matters: Revisiting intrinsic self-correction capabilities of large language models." arXiv preprint arXiv:2402.12563 (2024).
[3] Nie, Lunyiu, et al. "Online Cascade Learning for Efficient Inference over Streams." Forty-first International Conference on Machine Learning.

**Questions:**

Please see the weakness above.

---

> ### Author Response · Authors · 2024-11-20
> **Response to Reviewer cViA**
>
> We appreciate the reviewer’s positive feedback regarding the novelty of our idea (“very interesting”) and the promise of our results (“very promising”). Please find the answers to the questions below.
>
> > The writing of Section 2 could be clearer.
>
>
> We have addressed this by moving the related work to the appendix, and adding further details on our method (lines 235-277 in the revised pdf). We also added more motivations to Section 2.3. The changes are highlighted in blue.
>
> > The classification of this method is not very clear (intrinsic or extrinsic).
>
> According to [1], MAgICoRe is categorized as an extrinsic method. However, the only external feedback we utilize is step-wise scores, without any natural language feedback from a more powerful model. The rationale for using RM scores is that LLMs are not effective at identifying their own errors or self-correcting [2, 3]. Additionally, we do not rely on a large dataset of examples, whereas RAG would require a substantial corpus of solved problems to retrieve from. These problems would need to be similar to the test problem at hand.
>
> Moreover, it is not certain whether, even given this corpus and a strong retriever, models could in fact improve and refine their output. Doing so would involve understanding what commonalities the different retrieved solutions have and how these differ from the proposed solution, which is a challenging problem involving latent error localization. In contrast, PRM makes these problems explicit by directly identifying and localizing errors. If the reviewer is aware of any relevant works in this area, we would greatly appreciate any references.
>
> [1] https://arxiv.org/abs/2310.01798
>
> [2] https://arxiv.org/abs/2311.08516
>
> [3] https://arxiv.org/pdf/2406.01297
>
> > The framework is a combination of multi-agent and RM for selective refinement.
>
> The scope of our work is indeed to perform multi-agent refinement. Multi-agent systems have been widely applied to enhance faithfulness [4], reasoning [5], and translation [6], but there has been limited success in demonstrating their effectiveness for refinement tasks. Thus, while we acknowledge that MAgICoRE combines multi-agent systems with reward models for selective refinement, one of its key contributions lies in showcasing the effectiveness of multi-agent systems for LLM refinement.
>
> [4] https://arxiv.org/abs/2305.14325
>
> [5] https://arxiv.org/abs/2309.13007
>
> [6] https://arxiv.org/abs/2305.19118
>
> > SVAMP is a dataset with 700 training and 300 testing samples.
>
> We thank the reviewer for their careful attention. In SVAMP’s original paper (https://aclanthology.org/2021.naacl-main.168/) and code base (https://github.com/arkilpatel/SVAMP), we didn’t find that they have an official split for training and testing. Therefore, we follow previous work [7,8,9] to use all 1000 samples for evaluation, since we think a larger test set can reduce variations.
>
> [7] https://arxiv.org/abs/2205.11916
>
> [8] https://arxiv.org/abs/2309.17452
>
> [9] https://arxiv.org/abs/2309.05653
>
> > Please specify the subtasks in MMLU as used.
>
>
> We thank the reviewer for their careful attention. We are using the MMLU-Math subset curated by MAmmoTH [10]. We have added this in line 323.
>
> [10] https://arxiv.org/abs/2309.05653
>
> > Please add GPT4 for a subset of the test datasets.
>
> Following prior work [11,12,13] and considering the high cost of running GPT-4 across all datasets, we focus on Llama3 and GPT-3.5 as base models. However, **we conducted an additional experiment using GPT-4o-mini on the MATH 500 dataset**, primarily comparing 40-way SC with the weighted SC variation that incorporates PRM scores for vote weighting. Note that in Figure 4 in our submission, we found that 120-way SC actually decreases the performance on MATH. The results demonstrate that **MAgICoRE can also enhance GPT-4o-mini’s performance**, albeit with a smaller margin of improvement compared to Llama3-8B and GPT-3.5. We have added these results in lines 489-500.
>
> | Method               | Accuracy  |
> |----------------------|-----------|
> | Zero-shot            | 72.0      |
> | 40-way SC            | 79.2      |
> | 40-way SC + PRM      | 79.4      |
> | MAgICoRE Iter=1      | 80.2      |
> | MAgICoRE Iter=2      | **80.4**  |
> | MAgICoRE Iter=3      | **80.4**  |
>
>
> [11] https://arxiv.org/pdf/2305.14934
>
> [12] https://arxiv.org/pdf/2311.07961
>
> [13] https://arxiv.org/pdf/2305.14325
>
>
> > What is the cost of MAgICoRE for comparison?
>
> We include a token count comparison in **Figure 5** of the revised version. We find that scaling 40-way Self-Consistency (SC) to 120-way SC largely increases token usage but yields a limited performance improvement (and even a drop on MATH). In contrast, **MAgICoRE effectively improves the performance with a larger token budget**. Notably, MAgICoRE uses fewer tokens on datasets such as SVAMP, GSM8K, and SAT compared to 120-way SC, while achieving superior performance. These results are further described in lines 997-1020 in the revised pdf.

---

> > ### Comment · Reviewer_cViA · 2024-11-22
> >
> > I have read the rebuttal of the authors, and it addresses most of my concerns. One question I still have in mind is the last point. The authors stated that "We include a token count comparison in Figure 5 of the revised version." After reading the description of Figure 5, it seems like the PRM and ORM's tokens are not counted as part of the total cost? Please clarify if you also included the cost of PRM and ORM in the figure. If you did not include them and fear that including those tokens will have negative results on the figure, please clarify. You could also consider to use FLOPs as the cost metric.

---

> > > ### Author Response · Authors · 2024-11-22
> > > **Response to Reviewer cViA**
> > >
> > > Thanks for the continued engagement, and for your suggestions – we are glad we addressed most of your points. To answer your last question: **adding the ORM and PRM token costs does not change our positive conclusions in Figure 5\.** We have re-computed the token cost taking into account the ORM and PRM tokens. We have revised the pdf and included an updated version of Figure 5 with these new token counts. The high-level takeaways remain roughly the same: while SC stagnates with additional tokens, MAgICoRE improves as we add more tokens. Moreover, one iteration of MAgICoRE consistently outperforms 120-way SC while using fewer tokens. We have updated the pdf to reflect these results, and highlighted the changes in green. Given that we use both open-sourced and API-based models, we have opted to highlight the token cost as this is more informative for API-based cost estimation. We hope that this addresses your remaining question and – if so – we would appreciate if you could revisit your score.

---

> > > > ### Author Response · Authors · 2024-11-25
> > > > **Reminder to Reviewer cViA**
> > > >
> > > > Thanks again for your suggestions and for engaging in discussion with us. Since the rebuttal period is drawing to a close, with only 1 day left before the 26th, we wanted to check in again and see if our additional experiments (including an updated plot as per your latest suggestion) and our responses have addressed your comments. If they have, we would appreciate if you could revisit your score accordingly.

---

> ### Author Response · Authors · 2024-11-27
> **Reminder for Reviewer cViA**
>
> Thank you once again for your valuable feedback and suggestions. Since the PDF update window closes in 24 hours, we’d greatly appreciate your confirmation on whether our additional positive results—**MAgICoRE's improvement of GPT4o-mini and the detailed token cost comparison that shows its cost-effectiveness**—along with our responses, address your comments. If they do, we kindly ask you to consider revisiting your score.

---

> ### Author Response · Authors · 2024-11-30
> **Reminder for Reviewer cViA**
>
> Dear Reviewer cViA,
>
> As the deadline for the discussion period approaches, we wanted to follow up and see if our additional experiments and responses have addressed your comments. If they have, we would greatly appreciate it if you could revisit your score accordingly.
>
> Best,
>
> The Authors

---

### Official Review · Reviewer_eYto · 2024-11-03

**Soundness:** 3
**Presentation:** 2
**Contribution:** 2
**Rating:** 3
**Confidence:** 4

**Summary:**

The paper introduces MAGICORE, a multi-agent, iterative coarse-to-fine refinement framework designed to improve reasoning performance in LLMs. The system uses three agents—Solver, Reviewer, and Refiner—who collaborate iteratively, guided by external RMs that provide both global and step-wise feedback. The framework is evaluated on five math reasoning datasets, showing consistent gains across models and datasets, outperforming existing self-refinement and aggregation-based baselines.

**Strengths:**

MAGICORE combines a multi-agent system with a coarse-to-fine approach, allowing efficient resource allocation that avoids the pitfalls of excessive or insufficient refinement.

The integration of external reward models (ORM and PRM) for both global and step-wise scoring enables precise error identification and correction.

MAGICORE outperforms higher-budget baselines with fewer samples, making it a promising framework for cost-effective large-scale reasoning.

**Weaknesses:**

The three technical designs to address the three major challenges seem isolated and independent, making the overall framework not a “unified” solution.

Each of the three technical designs has been fully utilized by existing research works, e.g., least-to-most prompting, multi-agent discussion, and iterative refinement. This paper does not propose any fundamentally new method, limiting its technical contributions.

The selected baselines for comparison are insufficient. Several self-correction, self-refinement, tree searching, or multi-agent reasoning methods are not included.

The experiments are only conducted on math reasoning datasets, leaving several other domains unexplored, e.g., program coding, commonsense reasoning, and symbolic reasoning. These are also essential domains in LLM reasoning.

Only two LLMs are considered. Several up-to-date and more powerful models, e.g., GPT-4 and llama3 70B/405B, should also be investigated.

The paper seems too dense to read. The presentation necessitates much improvement.

**Questions:**

Is the MAGICORE framework suitable for other types of reasoning? What modifications would be necessary for other domains? For example, in terms of condition evaluation and the refinement process?

How does MAGICORE handle misleading questions where the Solver consistently arrives at the same incorrect answer across multiple solutions? Would MAGICORE classify such cases as 'easy,' given that only one of the ORM or PRM needs to be 'fooled' during classification?

The refinement process relies on PRM-generated step-wise scores to guide targeted feedback. How sensitive is this process to the accuracy of PRM's step-wise evaluations? If PRM incorrectly assigns high scores to flawed steps, could this lead to ineffective or even harmful refinements? How does MAGICORE mitigate such risks in error localization?

The reviewer and refiner roles are distinct, with the Reviewer generating feedback based on step-wise scores. What specific mechanisms ensure that the Reviewer’s feedback is both relevant and actionable for the Refiner? Are there cases where ambiguous or overly general feedback could hinder the refinement process?

---

> ### Author Response · Authors · 2024-11-20
> **Response to Reviewer eYto (Part 1)**
>
> >The three technical designs to address the three major challenges seem isolated
>
>
> While our framework does consist of multiple connected components to address the three interrelated issues, we would like to clarify that we are not solving each issue one by one, but instead, solving the three issues jointly through a communicative multi-agent system, as each module in our system re-uses components and relies on the other modules. That is:
> To address “Excessive refinement” issue, we adopt reward models (ORM and PRM) to categorize each instance as easy or hard, and only refine hard instances.
> To address “Inability to localize errors”, the PRM is re-used to obtain targeted feedback via the “reviewer” agent. This helps the reviewer LLM localize the error and generate helpful feedback.
> To address “Insufficient refinement”, we propose a multi-agent interactive refining framework that communicates between the reviewer and refiner and refines based on information from the same PRM and ORM to iteratively refine the output. This only applies to hard instances, and we re-use the ORM to evaluate whether the refinement enhances quality.
>
> Thus, the approach to the challenges is tied together by the use of the RMs and the communication between agents (the Solver, the Reviewer, and the Refiner).
>
> As per your suggestion, in addition to our existing ablations in Table 2 - Table 5 in the original pdf, **we have added an additional ablation study for each issue individually to clearly demonstrate the significance of addressing all issues jointly**. The setup and results are presented below:
>
> |                      Method                   | Selective Refinement | PRM Fine-Grained Feedback | Iterative Refinement |
> |-----------------------------------------------------|:--------------------:|:-------------------------:|:--------------------:|
> | Only Address Issue 1  (Excessive Refinement)        |           yes          |             no             |           no          |
> | Only Address Issue 2  (Inability to Localize Error) |           no          |             yes             |           no          |
> | Only Address Issue 3 (Insufficient Refinement)      |           no          |             no             |           yes          |
>
> | Method | MMLU | MATH |
> |---------|--------|------|
> | Only address issue 1 | 64.7 | 44.0 |
> | Only address issue 2 | 65.9 | 45.4 |
> | Only address issue 3 | 60.3 | 36.4 |
> | MAgICoRE (Iter=3)    | **68.9** | **47.8** |
>
> The results show that
> - “Only Address Issue 1” is less effective due to the absence of PRM-based targeted feedback and iterative refinement.
> - “Only Address Issue 2” incorporates PRM scores for all samples, but this sacrifices efficiency and still underperforms the full setup used in MAgICoRE.
> - “Only Address Issue 3” performs the worst, as it uniformly refines all samples without leveraging PRM-based targeted feedback.
>
> Thus, **all three issues must be addressed jointly, as MAGiCoRE does.** We have added these results and a discussion of them to lines 455-472 in the revised pdf.

---

> > ### Author Response · Authors · 2024-11-20
> > **Response to Reviewer eYto (Part 2)**
> >
> > > Comparison with least-to-most prompting, multi-agent discussion, and iterative refinement
> >
> > We would like to highlight that the main contribution of our work is demonstrating that multi-agent systems can effectively incorporate refinement. Nevertheless, **we compare against all these additional baselines, finding that MAgICoRE outperforms the best of them by 2.5% with only one iteration**.
> >
> > First, we compare to the Self-Correct RCI prompt used in [2] and also augment it with 120-way Self-Consistency to ensure fairness. We find that **MAgiCoRE outperforms Self-Correct by 6.1% with only one iteration**. While the Self-Correct prompt also leverages an LLM's intrinsic ability to identify and correct errors, our results indicate this approach is less effective than MAgiCoRE.
> >
> > Second, we compare to least-to-most prompting with 120-way Self-Consistency to ensure fairness, finding that **MAgiCoRE outperforms least-to-most by 3.9% with only one iteration**. Least-to-most prompting makes a distinction between easy and hard problems based on the number of subquestions associated with them, but it does not address refinement or its challenges (such as excessive refinement/iterative refinement) or the inability to localize errors.
> >
> > We also add a **multi-agent debate baseline, finding that MAgICoRE outperforms it by 2.5% with only one iteration**. While multi-agent debate demonstrates that discussion-based test-time aggregation improves performance, it is limited by the fact that agents must reach a consensus, which heavily relies on each model’s intrinsic abilities and the ability to persuade other agents.
> >
> > In our original version, we already compared with Self-Refine (lines 297-300 in the original pdf, 312-315 in the revised pdf), enhanced with 120-way self-consistency for a fair comparison to our existing baselines. Here, we find that **MAgICoRE outperforms Self-Refine by 4.0%**. Note that iterative refinement methods such as Self-Refine also fail to address the problem of excessive refinement and depend on an LLM’s inherent ability to identify and localize errors (which past work has called into question) [5].
> >
> > The full results are given in the following table and discussed further in lines 864-895 of our revised pdf:
> >
> > | Method | MMLU | MATH | SVAMP | GSM8K | SAT | Avg. |
> > |--------|-------|-------|-------|-------|-------|-------|
> > | 120-way SC | 63.0 | 40.6 | 89.8 | 90.3 | 70.5 | 70.8 |
> > | 120-way SC + PRM [1] | 65.4 | 44.6 | 90.8 | 90.7 | 72.5 | 72.8 |
> > | Self-correct + 120-way SC [2] | 62.1 | 38.6 | 86.2 | 88.1 | 65.6 | 68.1 |
> > | Least-to-Most + 120-way SC [3] | 62.6 | 40.6 | 89.0 | 90.3 | 68.9 | 70.3 |
> > | Multi-Agent Debate + SC [4] | 64.6 | 41.0 | 89.6 | 90.8 | 72.5 | 71.7 |
> > | MAGICORE (Iter=1) | 67.3 | 46.0 | **91.4** | 91.1 | 75.0 | 74.2 |
> > | MAGICORE (Iter=2) | 68.4 | 47.2 | 91.1 | **92.3** | 76.4 | 75.1 |
> > | MAGICORE (Iter=3) | **68.9** | **47.8** | 91.3 | 91.6 | **78.2** | **75.6** |
> >
> > [1] https://aclanthology.org/2023.acl-long.291/
> >
> > [2] https://arxiv.org/abs/2303.17491
> >
> > [3] https://arxiv.org/abs/2205.10625
> >
> > [4] https://arxiv.org/abs/2305.14325
> >
> > [5] https://arxiv.org/abs/2310.01798

---

> > > ### Author Response · Authors · 2024-11-20
> > > **Response to Reviewer eYto (Part 3)**
> > >
> > > > The experiments are only conducted on math reasoning datasets
> > >
> > > We primarily focus on math reasoning because of the current development of Process Reward Models (PRMs), which has been focused on math. We believe ongoing advancements in reward models will help overcome this bottleneck [6]. Moreover, PRMs are domain-specific models, and so creating a new PRM for a domain is a major challenge and likely a topic for one (or several) full papers.
> > >
> > > Nevertheless, based on your suggestion, **we add the following experiments on a commonsense reasoning task: ARC-Challenge [7], and a logical reasoning task (as categorized by [8]): Date Understanding**. We randomly sample 200 instances and use GPT4o-mini as a PRM for our experiments. Specifically, we prompt GPT4o-mini to provide step-wise correctness scores without any textual explanations or reasoning, acting the same as a PRM (indeed, we manually verified that no explanations were generated by manually evaluating 100 examples). This approach ensures that our Solver (Llama3-8B) does not have access to explanations from a stronger model (GPT4o-mini). Instead, the Reviewer (Llama3-8B) relies solely on the step-wise numerical scores from GPT4o-mini to generate feedback, and the Refiner (Llama3-8B) uses this self-generated feedback for improvement.
> > >
> > > Results below show that **MAgICoRE generalizes well on commonsense and logical reasoning tasks, outperforming 120-way SC by 2.5% on commonsense reasoning and 8.0 % on logical reasoning**. Full results and discussion can be found in lines 473-489 of our revised pdf.
> > >
> > >
> > > | Method              | ARC    | Date    |
> > > |---------------------|--------|---------|
> > > | Zero-shot           | 66.5   | 52.5    |
> > > | 40-way SC           | 85.5   | 72.5    |
> > > | 120-way SC          | 86.0   | 72.5    |
> > > | MAgICoRE Iter = 1   | 87.5   | 79.5    |
> > > | MAgICoRE Iter = 2   | 88.0   | 79.5    |
> > > | MAgICoRE Iter = 3   | **88.5** | **80.5** |
> > >
> > > [6] https://arxiv.org/abs/2403.13787
> > >
> > > [7] https://arxiv.org/abs/1803.05457
> > >
> > > [8] https://arxiv.org/abs/2206.04615
> > >
> > > > Only two LLMs are considered
> > >
> > > Following prior work [9,10,11] and considering the high cost of running GPT-4 across all datasets, the experiments in our submitted version focused on Llama3 and GPT-3.5 as base models. However, **we have added an additional experiment using GPT-4o-mini on the MATH 500 dataset**, primarily comparing 40-way SC with the weighted SC variation that incorporates PRM scores for vote weighting. Note that in Figure 4 (original and revised pdf), we found that 120-way SC actually decreases the performance on MATH, so we omit 120-way SC here. The results demonstrate that **MAgICoRE can also enhance GPT-4o-mini’s performance**, albeit with a smaller margin of improvement compared to Llama3-8B and GPT-3.5.  We have added these results in lines 489-500 of the revised pdf.
> > >
> > > | Method               | Accuracy  |
> > > |----------------------|-----------|
> > > | Zero-shot            | 72.0      |
> > > | 40-way SC            | 79.2      |
> > > | 40-way SC + PRM      | 79.4      |
> > > | MAgICoRE Iter=1      | 80.2      |
> > > | MAgICoRE Iter=2      | **80.4**  |
> > > | MAgICoRE Iter=3      | **80.4**  |
> > >
> > >
> > >
> > >
> > >
> > >
> > > [9] https://arxiv.org/pdf/2305.14934
> > >
> > > [10] https://arxiv.org/pdf/2311.07961
> > >
> > > [11] https://arxiv.org/pdf/2305.14325
> > >
> > >
> > > > How does MAgICoRE handle misleading questions where the Solver consistently arrives at the same incorrect answer across multiple solutions?
> > >
> > > We thank the reviewer for raising this question. First, the concepts of "easy" and "hard" in this work are used to indicate whether a problem requires refinement and do not necessarily reflect the intrinsic difficulty of the problem. However, as noted in Appendix E of our revised pdf, this classification aligns well with human judgment.
> > >
> > > Second, Self-Consistency has been shown to be effective across a wide range of tasks because aggregating multiple generations tends to cancel out noise. As a result, the likelihood of the Solver consistently arriving at the incorrect answer is low when we take multiple samples.

---

> > > > ### Author Response · Authors · 2024-11-20
> > > > **Response to Reviewer eYto (Part 4)**
> > > >
> > > > > How sensitive is this process to the accuracy of PRM's step-wise evaluations?
> > > >
> > > > **We present an additional experiment to assess the PRM’s accuracy** by evaluating the solution-level score (calculated as the product of step-wise scores) and comparing its predictions to ground truth answers. On the MATH dataset, the PRM achieves an F1 score of 58.0. For comparison, using randomly generated step-wise scores yields an F1 score of 31.5. Additionally, when PRM scores are randomly perturbed by ±25%, the F1 score decreases slightly to 57.1. The overall accuracy across these three setups on the MATH test set is summarized below:
> > > >
> > > > | Setting            | F1 Score | Overall Accuracy  |
> > > > |--------------------------|--------------|---------------------------|
> > > > | Random step-wise scores  | 31.5       | 43.8                      |
> > > > | Perturbed PRM scores     | 57.1       | 45.4                      |
> > > > | Actual PRM scores        | **58.0**        | **46.0**                      |
> > > >
> > > > Results show that the **PRM scores substantially outperform random scores and are resilient to moderate perturbations while maintaining overall accuracy.**
> > > >
> > > > > What specific mechanisms ensure that the Reviewer’s feedback is both relevant and actionable for the Refiner?
> > > >
> > > > We do not specifically filter the reviewer’s feedback before passing it to the refiner, but we have two strategies to ensure the feedback is useful (1) we have a 1-shot demonstration of how to incorporate step-wise PRM scores to generate targeted and actionable feedback, and (2) our multi-agent communication across iterations between the reviewer and refiner means that unhelpful feedback will result in lower scores and thus lead to continued refining. Qualitatively, we find that the reviewer provides both helpful and unhelpful feedback, as shown in the qualitative examples in Table 14-Table 17.
> > > >
> > > > We agree that there are cases when the feedback is not useful, but we would like to highlight that the main motivation of our “selective refinement”, i.e., deciding when to refine, is to reduce the cases when refinement hurts, and this is reflected in the performance. Moreover, the Reviewer (which generates feedback) is also an LLM, thus even given a correct error localization, it can come up with wrong feedback. For a detailed analysis, please refer to the ablation studies on applying refinement uniformly in Table 2 in the original pdf – it shows that without selective refinement, performance can substantially decline.
> > > >
> > > > > Paper density
> > > >
> > > > We appreciate the reviewer’s valuable feedback on the writing, and we made several improvements to enhance clarity: (1) moved related work to the appendix to relax the space limit. (2) added more method details and motivations to Section 2.3. (3) stressed that the three refinement issues are tied and that we are jointly addressing the three issues instead of independently. These modifications are highlighted in blue in the revised version.

---

> > > > > ### Author Response · Authors · 2024-11-23
> > > > > **Reminder to reviewer eYto**
> > > > >
> > > > > Thank you once again for your valuable feedback. We hope our response has addressed all of your questions and will allow you to revisit your score, otherwise we would be happy to engage further and address any further questions you might have in the remaining few days of the discussion period.

---

> > > > > > ### Author Response · Authors · 2024-11-25
> > > > > > **Reminder to reviewer eYto**
> > > > > >
> > > > > > Since the rebuttal period is drawing to a close, with only 1 day left before the 26th, we wanted to check in again and see if our additional positive results and our responses have addressed your comments. Based on your suggestions, we have shown positive results on transferring to two other domains as well as transfer to GPT4o-mini and have sought to address your other comments and questions in detail in our rebuttal. If your concerns have been addressed, we would appreciate if you could revisit your score accordingly.

---

> ### Author Response · Authors · 2024-11-27
> **Reminder for Reviewer eYto**
>
> Thank you once again for your valuable feedback and suggestions. Since the PDF update window closes in 24 hours, we’d greatly appreciate your confirmation on whether our additional positive results—**MAgICoRE's generalization to two other domains and its improvement of GPT4o-mini**—along with our responses, address your comments. If they do, we kindly ask you to consider revisiting your score.

---

> ### Author Response · Authors · 2024-11-30
> **Reminder for Reviewer eYto**
>
> Dear Reviewer eYto,
>
> As the deadline for the discussion period approaches, we wanted to follow up and see if our additional experiments and responses have addressed your comments. If they have, we would greatly appreciate it if you could revisit your score accordingly.
>
> Best,
>
> The Authors

---

### Official Review · Reviewer_k9ff · 2024-11-04

**Soundness:** 3
**Presentation:** 3
**Contribution:** 3
**Rating:** 6
**Confidence:** 4

**Summary:**

MAGICORE is a multi-agent framework designed to improve large language model (LLM) reasoning by selectively refining answers based on difficulty, leveraging external reward models (RMs) for targeted feedback and iterative refinement. This approach addresses key issues in traditional refinement methods, such as over-correction and insufficient error localization, achieving superior performance across five math reasoning datasets compared to baseline methods. Notably, MAGICORE shows continued improvement with more iterations, unlike other methods, highlighting the importance of selective and iterative refinement using RMs.

**Strengths:**

1. This paper proposes multi-agent framework, named MAGICORE, effectively categorizes problems into "easy" and "hard" cases, applying coarse-grained aggregation to simpler problems and fine-grained, multi-agent refinement to challenging ones. This selective refinement approach minimizes over-correction and ensures computational resources are allocated efficiently, resulting in higher performance without excessive sampling.

2. By incorporating step-wise Reward Model (RM) scores, MAGICORE significantly improves error localization, allowing for more accurate, step-by-step feedback. This targeted feedback mechanism enables the framework to address specific mistakes precisely, enhancing the overall quality of model outputs.

3. This paper conducts comprehensive experiments to verify the effectiveness of the proposed framework MAGICORE.

**Weaknesses:**

1.  MAGICORE relies heavily on external Reward Models to assess difficulty and provide targeted feedback, which could introduce dependency on the quality of these RMs. However, the better reward model is not easy to obtain. If the RMs are not well-tuned or suited to the specific dataset, the refinement process might misidentify errors, potentially impacting overall performance.

2. While MAGICORE shows strong results on math reasoning tasks, it is unclear how well it would perform on other types of reasoning tasks, which may have different error patterns. More evaluations should be conducted across a broader range of tasks to verify the generalizability of this framework beyond math.

3. Although MAGICORE improves with additional iterations, the optimal stopping point remains unclear, despite the authors conducting experiments with 1 to 5 iterations for validation.

**Questions:**

See Weaknesses

---

> ### Author Response · Authors · 2024-11-20
> **Response to Reviewer k9ff**
>
> We thank the reviewer for recognizing the effectiveness and efficiency of our method. Please find the answers to the questions below:
>
> > MAgICoRE relies heavily on external reward models
>
> We acknowledge that external reward models play an important role in our framework. While MAgICoRE does utilize external reward models, our framework is modular and can readily incorporate new reward models as they emerge. This gives MAgICoRE an advantage, as the community is actively improving reward models (as evidenced by the rapid rate of improvement on reward model benchmarks) [1]. Our approach is thus complementary to and enhanced by progress in reward modeling, rather than constrained by it.
>
> Furthermore, existing work has found that LLMs often struggle to recognize their own errors but can effectively correct them when the mistakes are explicitly identified [2]. External feedback on error localization is therefore critical for refinement to success. While it is possible to train a custom domain-specific error-identification model, this approach is often data-dependent and prone to obsolescence. In contrast, MAgICoRE's modular design overcomes this limitation by enabling the integration of new state-of-the-art models as they become available. Moreover, our results on generalizing to commonsense and logical reasoning show that even when RMs are unavailable, we can use sufficiently strong LLMs like GPT4o-mini in place of trained RMs. We have added this discussion to lines 514-524 in our revised pdf.
>
> [1] https://arxiv.org/abs/2403.13787
>
> [2] https://arxiv.org/abs/2311.08516
>
> > More evaluations should be conducted across a broader range of tasks
>
> We focus primarily on math reasoning due to the current development of the Process Reward Models (PRM), which has been more math-focused. We believe ongoing advancements in reward models will help overcome this bottleneck [5]. Moreover, PRMs are domain-specific models, so creating a new PRM for a domain is a major challenge and likely a topic for one (or several) full papers.
>
> Nevertheless, based on your suggestion, **we add the following experiments on a commonsense reasoning task: ARC-Challenge [3], and a logical reasoning task: Date Understanding [4]**. We randomly sample 200 instances and use GPT4o-mini as a PRM for our experiments. Specifically, we prompt GPT4o-mini to provide step-wise correctness scores without any textual explanations or reasoning, acting the same as a PRM (indeed, we manually verified that no explanations were generated by manually evaluating 100 examples). This approach ensures that our Solver (Llama3-8B) does not have access to explanations from a stronger model (GPT4o-mini). Instead, the Reviewer (Llama3-8B) relies solely on the step-wise numerical scores from GPT4o-mini to generate feedback, and the Refiner (Llama3-8B) uses this self-generated feedback for improvement.
>
> Results below show that **MAgICoRE generalizes well on commonsense and logical reasoning tasks, outperforming 120-way SC by 2.5% on commonsense reasoning and 8.0 % on logical reasoning**. Full results and discussion can be found in lines 473-489 of our revised pdf.
>
> | Method              | ARC    | Date    |
> |---------------------|--------|---------|
> | Zero-shot           | 66.5   | 52.5    |
> | 40-way SC           | 85.5   | 72.5    |
> | 120-way SC          | 86.0   | 72.5    |
> | MAgICoRE Iter = 1   | 87.5   | 79.5    |
> | MAgICoRE Iter = 2   | 88.0   | 79.5    |
> | MAgICoRE Iter = 3   | **88.5** | **80.5** |
>
> [3] https://arxiv.org/abs/1803.05457
>
> [4] https://arxiv.org/abs/2206.04615
>
> [5] https://arxiv.org/abs/2403.13787
>
>
> > The optimal stopping point is unclear
>
> We would like to point here to the results of 5 iterations shown in Figure 3 and Table 12 in the appendix (lines 350-363 in the original, lines 365-377 in the revised pdf). Our empirical findings suggest that 2 to 3 iterations generally yield the best performance. We would also like to note that the stopping criteria can flexibly align with budget constraints, as performance continues to improve with each iteration. For instance, on the MATH dataset, even if the process is stopped arbitrarily at iteration 3, (although the optimal number is 5 ) it would still surpass the performance of earlier iterations, such as iteration 1.

---

> ### Author Response · Authors · 2024-11-23
> **Reminder to reviewer k9ff**
>
> Thank you once again for your valuable feedback. We hope our response has addressed all of your questions and will allow you to revisit your score, otherwise we would be happy to engage further and address any further questions you might have in the remaining few days of the discussion period.

---

### Official Review · Reviewer_ywAv · 2024-11-04

**Soundness:** 3
**Presentation:** 3
**Contribution:** 2
**Rating:** 5
**Confidence:** 3

**Summary:**

This paper introduces MAGICORE, an inference framework for LLM reasoning that leverages a multi-agent approach with three agents: the Solver, the Reviewer, and the Refiner. The Solver generates multiple solutions, which are then scored by the Reviewer using a Process Reward Model. The Refiner subsequently refines solutions based on their quality and confidence scores. MAGICORE demonstrates superior performance over several aggregation methods.

**Strengths:**

1. The writing is clear and easy to follow.
2. The proposed method is efficient and straightforward, achieving better performance than self-aggregation and self-refinement methods.

**Weaknesses:**

1. The paper lacks a comparison with methods that also use a PRM. While baselines are based on self-aggregation and self-refinement, it remains unclear how MAGICORE would perform compared to methods like self-consistency with PRM. This omission may make the comparison in Figure 1 less conclusive.

2. The method assumes that LLMs can improve through fine-grained stepwise refinement. Adding preliminary experiments or a discussion section to validate this assumption could enhance the paper’s robustness.

**Questions:**

See weakness.

---

> ### Author Response · Authors · 2024-11-20
> **Response to Reviewer ywAv**
>
> We thank the reviewer for highlighting that “the writing is clear” and “the method is efficient and effective”. We provide detailed answers to each question below.
>
> > comparison with methods that also use a PRM.
>
> Thanks for this suggestion – we have added new experiments using Self-Consistency with PRM scores. Specifically, based on the approach taken by [1], we perform weighted Self-Consistency where the weight for a reasoning chain is the product of the step scores. While this baseline underperforms MAgICoRE, we would like to highlight that it is also inefficient as it uniformly generates 120 samples per question with the PRM scores (MAgICoRE is adaptive, whereas this baseline is not). The results are as follows:
>
> | Method                     | MMLU | MATH | SVAMP | GSM8K | SAT | Avg. |
> |----------------------------|----------|-----------|--------|----------|---------|----------|
> | 120-way SC                | 63.0     | 40.6      | 89.8   | 90.3     | 70.5    | 70.8     |
> | 120-way SC + PRM [1]      | 65.4     | 44.6      | 90.8   | 90.7     | 72.5    | 72.8     |
> | MAGICORE (Iter=1)         | 67.3     | 46.0      | **91.4**   | 91.1     | 75.0    | 74.2     |
> | MAGICORE (Iter=2)         | 68.4     | 47.2      | 91.1   | **92.3**     | 76.4    | 75.1     |
> | MAGICORE (Iter=3)         | **68.9**     | **47.8**      | 91.3   | 91.6     | **78.2**    | **75.6**     |
>
> We find that using PRM scores as aggregation does improve the performance over 120-way SC alone, but underperforms our method by 2.8% on average, underscoring the importance of *using PRM to generate targeted feedback*, and using that feedback for fine-grained refinement. We have added this in lines 864-895 of our revised pdf.
>
> > The method assumes that LLMs can improve through fine-grained stepwise refinement.
>
> We thank the reviewer for raising this important point. Our method indeed assumes that LLMs can improve with fine-grained, stepwise refinement, which we believe is backed by our empirical results. In Table 3, we compare the effectiveness of feedback generated from random versus PRM-predicted scores, showing that feedback with PRM scores leads to higher performance on MMLU (66.4 vs. 67.3) and MATH (43.8 vs. 46.0), suggesting that fine-grained PRM feedback enables effective LLM refinement. We also agree that further testing is valuable, so **we conducted additional oracle experiments to compare the role of the PRM score for refinement**.
>
> Specifically, we sample 500 instances from the Math-Shepherd dataset [2], which includes gold label correctness for each step. We evaluated four conditions: (1) No feedback, where the LLM self-refines; (2) Random PRM score, where feedback is generated from random PRM scores; (3) PRM predicted score, where feedback is based on predicted PRM scores; and (4) Oracle PRM score, where feedback uses the gold correctness labels.
>
> | Setting                      | Accuracy  |
> |-----------------------------|--------|
> | No feedback (LLM self-refine) | 48.30 |
> | Random PRM score            | 49.60 |
> | PRM predicted score         | 51.20 |
> | Oracle PRM score            | **52.40** |
>
> Aligned with Section 4.2 and Table 3, “PRM predicted score” outperforms “No feedback” and “Random PRM score”. Most importantly, **using oracle PRM scores achieves the highest performance. This indicates that given reliable stepwise scores, LLMs can effectively refine their solutions and improve**, and these findings are also supported by [3]. We further describe these results in lines 906-915 of our revised pdf.
>
>
> [1] https://aclanthology.org/2023.acl-long.291/
>
> [2] https://huggingface.co/datasets/peiyi9979/Math-Shepherd
>
> [3] https://arxiv.org/pdf/2311.08516

---

> > ### Author Response · Authors · 2024-11-23
> > **Reminder for reviewer ywAv**
> >
> > Thank you once again for your valuable feedback. We hope our response has addressed all of your questions and will allow you to revisit your score, otherwise we would be happy to engage further and address any further questions you might have in the remaining few days of the discussion period.

---

> > > ### Author Response · Authors · 2024-11-25
> > > **Reminder to reviewer ywAv**
> > >
> > > Since the rebuttal period is drawing to a close, with only 1 day left before the 26th, we wanted to check in again and see if our additional experiments and responses have addressed your comments. If they have, we would appreciate if you could revisit your score accordingly.

---

> ### Author Response · Authors · 2024-11-27
> **Reminder for reviewer ywAv**
>
> Thank you once again for your valuable feedback and suggestions. Since the PDF update window closes in 24 hours, we’d greatly appreciate your confirmation on whether our additional positive result—**MAgICoRE's improvement over a strong PRM-based baseline**—along with our responses, address your comments. If they do, we kindly ask you to consider revisiting your score.

---

> ### Author Response · Authors · 2024-11-30
> **Reminder for reviewer ywAv**
>
> Dear Reviewer ywAv,
>
> As the deadline for the discussion period approaches, we wanted to follow up and see if our additional experiments and responses have addressed your comments. If they have, we would greatly appreciate it if you could revisit your score accordingly.
>
> Best,
>
> The Authors

---

### Author Response · Authors · 2024-11-20
**General Response**

We thank the reviewers for their insightful comments and feedback. The reviewers recognized MAgICoRE as a **promising idea** (R4), demonstrating both **effectiveness and efficiency** (R1, R2, R3) through **comprehensive experiments** (R2), and **achieving state-of-the-art performance** (R5). We are also grateful for the acknowledgment that the presentation is **clear and easy to follow** (R1, R5).

Previous research indicates that LLMs have a limited intrinsic ability to identify and correct their own errors [1,2]. To address this, we incorporate reward models as an external source of feedback, which enhances the effectiveness of the refinement process.

We would like to emphasize that the primary contribution of MAgICoRE lies in its promising results in multi-agent refinement, which *effectively and jointly address* the three key challenges in LLM refinement: excessive refinement, error localization, and insufficient refinement. *Reward models play a critical role in tackling these issues, with the same local and global rewards being re-used by different agents in our multi-agent refinement framework.*

(1) Selective Refinement: Reward models enable MAgICoRE to avoid excessive refinement by only performing fine-grained refinement on the hard instances.

(2) Error Localization: PRMs provide step-wise scoring, facilitating targeted feedback generation that helps the LLM identify and localize errors, thereby improving refinement effectiveness.

(3) Iterative Improvement: They act as an improvement checkpoint within the iterative refinement loop, ensuring only improved solutions are retained, thereby reducing the risk of under-refinement.

We have updated our submission to incorporate the reviewers' suggestions and have detailed the specific adjustments made in response to their feedback below.

[1] https://arxiv.org/abs/2310.01798

[2] https://arxiv.org/abs/2311.08516

---

### Meta-Review · Area_Chair_qdaS · 2024-12-13

**Metareview:**

This paper introduces a framework for Multi-Agent Iterative Coarse-to-Fine Refinement, named MAgICoRe. Some reviewers found the presentation lacking, particularly in the clarity of methodological details. Additionally, the ablation studies were deemed insufficient, and the comparisons failed to include some SOTA methods. In conclusion, while the paper shows some promise, it requires significant improvements and the current version is not ready for acceptance.

**Additional Comments On Reviewer Discussion:**

The quality of this submission falls below the bar expected for ICLR. Four out of five reviewers have recommended rejection.

---

### Decision · Program_Chairs · 2025-01-22

Reject